# Time evolution of effective central charge and signatures of RG irreversibility after a quantum quench

**Axel Cortés Cubero**[⋆]

Institute for Theoretical Physics, Center for Extreme Matter and Emergent Phenomena,
Utrecht University, Princetonplein 5, 3584 CC Utrecht, The Netherlands

⋆ a.cortescubero@uu.nl

## Abstract

At thermal equilibrium, the concept of effective central charge for massive deformations of two-dimensional conformal field theories (CFT) is well understood, and can be defined by comparing the partition function of the massive model to that of a CFT. This temperature-dependent effective charge interpolates monotonically between the central charge values corresponding to the IR and UV fixed points at low and high temperatures, respectively. We propose a non-equilibrium, time-dependent generalization of the effective central charge for integrable models after a quantum quench, $c_{\text{eff}}(t)$, obtained by comparing the return amplitude to that of a CFT quench. We study this proposal for a large mass quench of a free boson, where the effective charge is seen to interpolate between $c_{\text{eff}} = 0$ at $t = 0$, and $c_{\text{eff}} \sim 1$ at $t \to \infty$, as is expected. We use our effective charge to define an "Ising to Tricritical Ising" quench protocol, where the charge evolves from $c_{\text{eff}} = 1/2$ at $t = 0$, to $c_{\text{eff}} = 7/10$ at $t \to \infty$, the corresponding values of the first two unitary minimal CFT models. We then argue that the inverse "Tricritical Ising to Ising" quench is impossible with our methods. These conclusions can be generalized for quenches between any two adjacent unitary minimal CFT models. We finally study a large mass quench into the "staircase model" (sinh-Gordon with a particular complex coupling). At short times after the quench, the effective central charge increases in a discrete "staircase" structure, where the values of the charge at the steps can be computed in terms of the central charges of unitary minimal CFT models. When the initial state is a pure state, one always finds that $c_{\text{eff}}(t \to \infty) \geq c_{\text{eff}}(t = 0)$, though $c_{\text{eff}}(t)$, generally oscillates at finite times. We explore how this constraint may be related to RG flow irreversibility.

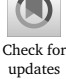
# 1   Introduction

Renormalization group (RG) transformations in quantum field theory are irreversible, as information about high-energy degrees of freedom is lost. This statement was formalized in (1+1)-dimensional field theories by A. B. Zamolodchikov [1], by showing that there always exists some function, $c(\{g\}, \mu)$, which decreases monotonically under RG flow (as the energy scale, $\mu$, is reduced), where $\{g\}$ are the coupling constants of the model. This function becomes stationary in a conformal field theory (CFT), where it can be shown to reduce to the corresponding central charge. For non-conformal theories, the function $c(\{g\}, \mu)$ can therefore be thought of as an effective central charge.

Such a $c$-function can be defined and is analytically tractable in the context of the thermodynamics of massive integrable deformations of CFT's [2], where it can be computed exactly through the tools of the thermodynamic Bethe ansatz (TBA) [3]. The TBA provides a formalism to compute exactly the partition function of an integrable field theory at a finite temperature, in the thermodynamic limit. The temperature-dependent effective central charge can be defined by comparing this exact partition function with the analogous partition function of a CFT at the same temperature, as will be reviewed in the following section. It is then easy to see that this function monotonically decreases as the temperature (energy scale) is reduced. Furthermore, this function smoothly interpolates between the values of the central charge of the CFT's describing the UV and IR dynamics, at high and low temperatures, respectively.

Our main objective is to define a non-equilibrium generalization of the effective central charge, which can be time-dependent. If such a function can be defined, we are interested in finding if there are any constraints on it which may be connected to the irreversibility of RG flow, which would be analogous to the constraints on the thermal $c$-function.

One particular non-equilibrium protocol that has been extensively studied in recent years is the so-called quantum quench. A quantum quench consists on initially preparing a system in an eigenstate of some Hamiltonian, $\mathcal{H}_0$, (typically the ground state), and then suddenly changing some parameter in the Hamiltonian, time-evolving the system unitarily with a new Hamiltonian, $\mathcal{H}$, with respect to which the system is no longer in an eigenstate. Such scenarios have been realized experimentally in cold atomic systems [4].

Quantum quenches generally introduce an extensive amount of energy into the system. If there is some type of equilibration at late times, this final state will therefore be described by some effective finite temperature (or a set of "effective temperatures"), whereas the initial state as we have described is a single pure eigenstate, which can be described as a zero-temperature state. The final state is in fact described by a generalized Gibbs ensemble (GGE) [5], which takes into account nontrivial conserved quantities besides the total energy, as we will mention in more detail in a later section. For integrable field theories, the GGE description amounts to performing thermal-like averages of observables, but with a different effective temperature for each momentum mode. For certain scenarios we will consider, in the limit of very large quenches (which introduce a very large amount of energy), it can be seen that the field theories thermalize, *i.e.* the effective temperature becomes a constant value for all momenta.

A sensible definition of effective central charge after a quantum quench should then interpolate between the central charge value which describes the zero-temperature dynamics of the initial state, and the finite temperature(s) effective central charge of the post-quench theory, as it evolves from $t = 0$, to $t \to \infty$. In particular, for large quantum quenches which lead to a thermal state at late times, one expects the effective central charge at late times to reproduce the known thermal value. Naturally, it is interesting to ask how exactly would such an effective charge evolve at finite times, as it interpolates between the two values. In particular, is the interpolation monotonically increasing, or does this non-equilibrium $c$-function oscillate?

In this paper, we propose a definition for effective central charge after a quantum quench, based on the so-called return amplitude, defined by the overlap between the state of the system at $t = 0$ and the time-evolved state.

The return amplitude can be computed exactly for quenches into a CFT (starting from the ground state of a massive theory), and can be expressed in terms of the effective temperature of the final state, and the corresponding central charge. This type of massive theory-to-CFT quenches can be thought of as a very large quench limit, and are thus known to thermalize at late times; only the overall effective temperature is included in the resulting GGE at late times. The time-dependent effective central charge in a non-conformal field theory is then defined by comparing the return-amplitude with that of a CFT quench, as we show in detail in Section 4. This definition is analogous to how the effective central charge at thermal equilibrium is defined, by comparing the partition function of the model to that of a CFT.

The return amplitude can be computed analytically in a class of quantum quenches of integrable field theories, where the initial state corresponds to integrable boundary conditions, as we discuss in Section 5. From such exact solutions, we can verify that our proposal for effective central charge seems to satisfy the expected properties, of interpolating between the central charges describing the initial and final states, particularly for large quenches, where the final central charge is the known thermal value. For smaller quantum quenches, where the late-time dynamics is not thermal, but described by a GGE, our proposal provides a possible definition of the concept of effective central charge corresponding to a GGE steady state.

As a simple example, we first consider the case of a mass quench of a free massive boson (where at $t = 0$, the boson mass is suddenly changed form $m_0$, to $m$. For large mass difference, $m_0 \gg m$, the quench introduces a large amount of energy, so at late times, we expect the system to be described by the UV central charge value $c = 1$, while the initial state is described by the IR value of $c = 0$. We find that our proposal for time-dependent effective central charge, indeed interpolates between these two values. Furthermore, we find that this function generally oscillates at finite times.

With our definition of effective central charge, we are able to define the notion of an "Ising to tricritical Ising" quantum quench, where at $t = 0$, the system is described by Ising field theory dynamics, corresponding to $c = 1/2$, and at late times, $t \to \infty$, the system is described by the tricritical Ising model with $c = 7/10$. As we show in Section 7, such a quench is obtained

by considering a specific deformation of the tricritical Ising CFT, which describes the massless RG flow between the two CFT's [6]. The Ising to tricritical Ising quench protocol consists on suddenly changing the coupling constant for this deformation. Interestingly, we find that this quench protocol can only be performed in one direction, *i.e.*, the reverse, "tricritical Ising to Ising" quench is impossible. This is simply because the tricritical Ising point describes the UV dynamics, and a quantum quench cannot remove enough energy from the system, for the final state to be described by the IR, Ising dynamics.

Similar quench protocols can be defined, which interpolate between any two adjacent unitary minimal CFT models at $t = 0$ and $t \to \infty$. It is interesting to notice the fact that these quenches can only be performed in one direction, going from a lower to a higher central charge. We propose that this strict direction in which quenches can be performed, is in some way connected to the irreversibility of RG flow, since this arises from the known properties of the equilibrium $c$-function, known to be related to RG flow.

We finally study quantum quenches into the so-called "staircase model" in Section 8. This model is defined as a specific analytic continuation of the sinh-Gordon model. At high temperatures, the effective central charge has been shown to reach a series of plateaus, which resemble a staircase [7], and the values of the central charge at these plateaus correspond to the central charges of all the unitary minimal CFT models. We find that the non-equilibrium effective central charge also evolves at very short times with a "staircase" structure, where it increases in discrete steps, whose values are determined in terms of the central charges of minimal models.

We will observe that in all our examples, the effective central charge at late times has always to be larger or equal than the effective central charge at $t = 0$, when the initial state is a pure state. We propose that this seems to be a consequence of the irreversibility of the RG flow in the non-equilibrium time evolution. This proposal is natural given that after a quantum quench starting from a pure state, one will end up probing higher energy scales than those probed for $t < 0$. Nevertheless, the effective central charge generally oscillates at *finite* times, so the increase is not monotonic, this makes it difficult to find a direct RG flow interpretation for the meaning of the effective charge at finite times.

## 2 Effective central charge for integrable models at thermal equilibrium

We consider the thermodynamics of a CFT with central charge $c$. This is done by placing the theory on a Euclidean toroidal geometry, with periodic boundary in both directions, and denoting the lengths of the two dimensions as $L$ and $R$. Eventually, in the thermodynamic limit, we will take $L \to \infty$, keeping $R$ finite, which yields a cylindrical geometry, as pictured in Figure 1.a.

There are two ways of quantizing the theory on a cylinder, which correspond to considering the compact dimension of length $R$ to be either the temporal or the spatial dimension. These are called $R$-channel and $L$-channel quantization, respectively. In the $R$-channel quantization, the length $R$ can be interpreted as the inverse temperature, $R = 1/T$. Therefore the cylindrical geometry yields the thermodynamics of the CFT. In the $L$-channel quantization, one considers instead the zero-temperature dynamics of the theory in a finite volume, $R$. By modular invariance of the CFT, both quantization procedures should yield equivalent results.

The partition function of the CFT in the $L$- and $R$-channels can be written as

$$Z(R, L) = \operatorname{Tr} e^{-L\mathcal{H}_R}, \tag{1}$$

$$Z(R, L) = \operatorname{Tr} e^{-R\mathcal{H}_L}, \tag{2}$$

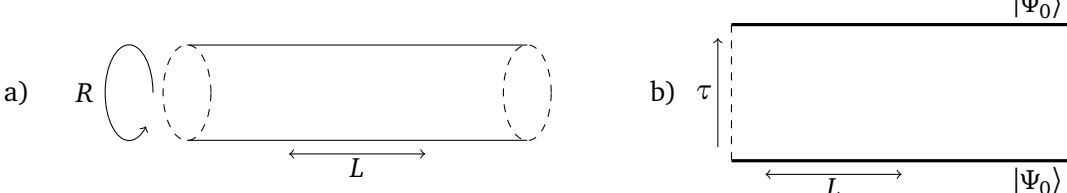

Figure 1: a) The thermal partition function corresponds to cylindrical geometry, with the circumference given by the inverse temperatue $R = 1/T$, and the system size, $L$, goes to infinity in the thermodynamic limit. b) The return amplitude for a quantum quench at imaginary times corresponds to computing the partition function on the strip geometry, with $\tau = it$, where the boundaries are given by the initial state.

respectively, where $\mathcal{H}_{R,L}$ are the Hamiltonians of the system quantized along the $R, L$ axis, and the trace is defined as a sum over the eigenstates of the corresponding Hamiltonian.

In the $L$-channel (1), taking the limit $L \to \infty$, implies that one only needs to consider the contribution from the ground state energy, $E_0(R)$ of the finite-volume Hamiltonian $\mathcal{H}_R$, therefore

$$Z(R, L) \approx e^{-LE_0(R)}. \tag{3}$$

The limit $L \to \infty$ in the $R$-channel (2) amounts to considering the thermodynamic limit of the theory at finite temperature $T = 1/R$. This means the partition function can be expressed as

$$Z(R, L) \approx e^{-LRf(R)}, \tag{4}$$

where $f(R)$ is the free energy per unit length. By comparing the two expressions for the partition function, we have $E_0(R) = Rf(R)$.

A CFT has no dimensionful parameters, so by dimensional analysis, the ground state energy needs to be of the form $E_0(R) = \text{const}/R$. The exact expression for the ground state energy is well known [8], and we only cite the result here:

$$E_0(R) = \frac{2\pi}{R}\left(\Delta_{min} + \bar{\Delta}_{min} - \frac{c}{12}\right) \equiv -\frac{\pi c_{\text{eff}}}{6R}, \tag{5}$$

where $\Delta_{min}, \bar{\Delta}_{min}$ are the minimum conformal weights of primary operators of the CFT. For a unitary CFT (which for simplicity, is the only kind we will consider in this paper), $\Delta_{min} = \bar{\Delta}_{min} = 0$, such that $c_{\text{eff}} = c$.

We can now consider the thermodynamics of a non-conformal integrable field theory (IFT) with some intrinsic mass scale $M$. In particular, one can consider a field theory which is an integrable deformation of a CFT, whose action is given by

$$S_{IFT} = S_{CFT} + \lambda \int \Phi(x)d^2x, \tag{6}$$

where $\Phi(x)$ is a relevant field of the CFT, and $\lambda$ is some dimensionful constant, which will be related to the mass scale $M$. At high energies, the dynamics of the model (6) are expected to be described effectively by the underlying CFT, $S_{CFT}$. Particularly, if one considers the thermal partition function of the integrable theory, at high temperatures one expects the partition function to be described by (5) with the appropiate $c_{UV}$ corresponding to $S_{CFT}$.

One can define a temperature-dependent effective central charge for an IFT at thermal equilibrium, simply by comparing the partition function to that of a CFT at the same temperature. In the IFT, the combination $MR$ is dimensionless, which means that the corresponding ground state energy $E_0(R)$ can be in general much more complicated function of $R$ than the expression (5), nevertheless, it is expected to approach (5) for $R \to 0$. An effective thermal central charge can be defined as

$$c_{\text{thermal}}(R) = \frac{6R}{\pi L} \log Z(R, L).$$ (7)

If the IFT (6) can be described at low energies by an infrared fixed point with central charge $c_{IR}$, it can be shown that $c_{\text{thermal}}(R)$ is a function that interpolates between $c_{IR}$ at $R \to \infty$ and $c_{UV}$ at $R \to 0$. We point out that for an integrable theory of massive particles, $c_{IR} = 0$.

The function $c_{\text{thermal}}(R)$ can be computed for an IFT through the thermodynamic Bethe ansatz formalism. The TBA program allows one to compute the partition function of an IFT given as an input the theory's two-particle S-matrix. We will not show the details of the derivation of the partition function, which can be found in [3], but only cite the necessary results. For now we will consider only IFT's with one species of particle, for simplicity.

The energy and momentum of a particle of mass, $m$, in an IFT can be parametrized as

$$E = m \cosh \theta, \qquad p = m \sinh \theta,$$

respectively, where $\theta$ is the particle's rapidity. One can define particle creation and annihilation operators $A^\dagger(\theta)$, and $A(\theta)$. The zero-temperature ground state is defined by

$$A(\theta)|0\rangle = 0,$$

and multiparticle states can be defined as

$$|\theta_1, \ldots, \theta_n\rangle = A^\dagger(\theta_1) \ldots A^\dagger(\theta_n)|0\rangle.$$

We denote the two-particle S-matrix as $S(\theta)$, such that

$$A^\dagger(\theta_1) A^\dagger(\theta_2) = S(\theta_1 - \theta_2) A^\dagger(\theta_2) A^\dagger(\theta_1).$$

The main result of the TBA formalism is that, given an S-matrix, $S(\theta)$, one can compute the partition function of an IFT in the thermodynamic limit, which is given by [3]

$$Z(L, R) = \exp\left[ \pm L \int \frac{d\theta}{2\pi} m \cosh \theta \log\left(1 \pm e^{-\varepsilon(\theta)}\right)\right],$$

where $\varepsilon(\theta)$ is the solution of the integral equation

$$\varepsilon(\theta) = mR \cosh \theta \mp \int \frac{d\theta}{2\pi} \varphi(\theta - \theta') \log\left(1 \pm e^{-\varepsilon(\theta')}\right),$$ (8)

where

$$\varphi(\theta) = -i \frac{d}{d\theta} \ln S(\theta),$$

and the $\pm$ signs are chosen to agree with the sign of $-S(0)$. The function $\varepsilon(\theta)$ is typically called the "pseudo energy". The effective central charge of an IFT at finite temperature is then given by

$$c_{\text{thermal}}(R) = \pm \frac{3}{\pi^2} mR \int d\theta \cosh \theta \log\left(1 \pm e^{-\varepsilon(\theta)}\right).$$ (9)

One simple example one can consider is the theory of a free massive boson, with $S(\theta) = 1$. In this case one expects the UV fixed point to be given by the CFT with $c_{UV} = 1$, and at low energies, $c_{IR} = 0$. Therefore $c_{\text{thermal}}(R)$ should interpolate between the values of 0 and 1. In this case, $\varphi(\theta) = 0$, such that

$$
\begin{aligned}
c_{\text{thermal}}(R) &= -\frac{3}{\pi^2} mR \int d\theta \cosh\theta \log\left(1 - e^{-mR\cosh\theta}\right) \\
&= \frac{6}{\pi^2} mR \sum_{n=1}^{\infty} \frac{1}{n} K_1(nmR),
\end{aligned}
\tag{10}
$$

where $K_\alpha(z)$ are modified Bessel functions. One can easily see, in the limit $mR \to 0$,

$$
c_{\text{thermal}}(0) = \frac{6}{\pi^2} \sum_{n=1}^{\infty} \frac{1}{n^2} = 1.
$$

In the $R \to \infty$ limit, we can use the asymptotic expression for the Bessel function

$$
K_\alpha(z) \sim \sqrt{\frac{\pi}{2z}} e^{-z} + \dots,
\tag{11}
$$

for $|\arg z| < \frac{3\pi}{2}$, and $|z| \to \infty$, to see that

$$
c_{\text{thermal}}(\infty) = 0,
$$

which confirms the expectations of the $c_{\text{thermal}}(R)$ function.

## 3   Quantum quenches and the return amplitude

In the quantum quench protocol, we consider a system that is initially prepared to be in a state $|\Psi_0\rangle$ that is an eigenstate (typically the ground state) of the pre-quench Hamiltonian, $H_0$. At time $t = 0$, the Hamiltonian is suddenly changed to $H$, for which $|\Psi_0\rangle$ is no longer an eigenstate. This state is then evolved unitarily as

$$
|\Psi_t\rangle = e^{-iHt}|\Psi_0\rangle.
$$

Some interesting quantities to compute are equal-time correlation functions of local operators $\Phi_i(x)$, defined as

$$
\frac{\langle\Psi_0|e^{iHt}\Phi_1(x_1)\Phi_2(x_2)\dots\Phi_n(x_n)e^{-iHt}|\Psi_0\rangle}{\langle\Psi_0|\Psi_0\rangle}.
$$

At long times, such quantities typically relax and it is expected that they can be described by a generalized Gibbs ensemble (GGE) [5], such that

$$
\lim_{t\to\infty} \frac{\langle\Psi_0|e^{iHt}\Phi_1(x_1)\Phi_2(x_2)\dots\Phi_n(x_n)e^{-iHt}|\Psi_0\rangle}{\langle\Psi_0|\Psi_0\rangle} = \text{Tr}\left[\Phi_1(x_1)\Phi_2(x_2)\dots\Phi_n(x_n)\rho_{GGE}\right],
$$

where

$$
\rho_{GGE} = \frac{e^{-\sum_i \beta_i Q_i}}{Z}, \qquad Z = \text{Tr} e^{-\sum_i \beta_i Q_i},
$$

and $Q_i$ are local[1] conserved charges of the theory. We note that if we include only the Hamiltonian, and no other conserved charges in the GGE, we recover the standard thermal ensemble.

The main quantity we will be interested in is the so-called return amplitude, defined as

$$\mathcal{F}(t) = \left| \frac{\langle \Psi_0 | e^{-iHt} | \Psi_0 \rangle}{\langle \Psi_0 | \Psi_0 \rangle} \right|. \tag{12}$$

This quantity gives a measure of how different is the time-evolved state, $|\Psi_t\rangle$ to the initial state $|\Psi_0\rangle$.

It is interesting to note that the return amplitude at imaginary values of time, $\tau = it$,

$$Z(\tau) = \mathcal{F}(-i\tau),$$

is the partition function of the theory on a strip geometry, in the crossed channel, where the roles of space and time are reversed. The inital state, $|\Psi_0\rangle$ now corresponds to the boundary conditions at the edge of the strip, as pictured in Figure 1.b. The system size, $L$, plays the role of the inverse temperature, and the the imaginary time, $\tau$ plays the role of the system size in the crossed channel. The identification between the return amplitude and the partition function in the crossed channel has some practical applications. For instance, the return amplitude may be computed in the crossed channel using the tools of the boundary thermodynamic Bethe ansatz [17]. One practical application of the computation of return amplitudes is the study of dynamical phase transitions [12], where phase transitions are identified at critical values of time, $t$, by searching for non analyticities in $\log[\mathcal{F}(t)]$, in analogy to how phase transitions at equilibrium can be found by studying non analyticities in the thermal partition function.

The partition function on the strip in the crossed channel, in the large-$L$, limit can generally be expressed as

$$Z(\tau) = \exp(-Lf(\tau)), \tag{13}$$

where $f(\tau)$ is the free energy in the boundary problem. This free energy can be split into three contributions, according to their behavior at large $\tau$:

$$f(\tau) = f_b \tau + 2f_s + f_C(\tau), \tag{14}$$

where $f_b$ and $f_s$ are bulk and surface energy contributions, and $f_C(\tau)$ decays at large $\tau$. The bulk term $f_b$ does not contribute to the return amplitude for real times, so we will not discuss it further. The surface term, $2f_s$ is related to the normalization of the initial state. For simplicity, we will choose $f_s$ such that the initial state is normalized as $\langle \Psi_0 | \Psi_0 \rangle = 1$, which means we will not need the normalization in the denominator in (12). This normalization implies the relation $2f_s = -f_C(0)$.

In this paper, the analogies between the return amplitude and the thermal partition function will be further exploited to define the concept of a time-dependent effective central charge, $c_{\text{eff}}(t)$ for quantum quench problems, based on the return amplitude for CFT quenches.

## 4    Return amplitude and central charge in CFT quenches

Quantum quenches where the post-quench Hamiltonian, $H$, describes a CFT have been extensively studied in a series of papers by Calabrese and Cardy [11]. In their approach, correlation

---

[1]It has recently been shown that to properly describe correlation functions after a quantum quench, sometimes it is necessary to relax the notion of locality, and that one must also consider certain quasilocal conserved charges, defined for quantum spin chains in [9], and for integrable field theories in [10]

functions of local fields are computed by considering CFT thermodynamics in a Euclidean strip geometry, and analytically continuing the results to real time. The simplest boundaries for the strip are those which preserve conformal invariance. If the pre-quench Hamiltonian $H_0$ describes a theory of particles with mass $m_0$, it was argued the initial state can be approximated by

$$|\Psi_0\rangle = e^{-\tau_0 H}|\Psi_0^*\rangle, \tag{15}$$

where $\tau_0 \sim m_0^{-1}$, is called the extrapolation length, and $|\Psi_0^*\rangle$ is a state that corresponds to conformally-invariant Dirichlet boundary conditions. The reason an extrapolation length is needed is that the state $|\Psi_0^*\rangle$ is not normalizable, and $\tau_0$ acts as a UV regulator.

The initial states (15) are in the majority of cases an oversimplification of the real initial states corresponding to a physical quantum quench. Nevertheless, considering such states can be a practical starting point for studying CFT quenches, since they allow for simple analytic computations. As we will discuss later, there are some regions of the parameter space where for some quenches, the simple initial states (15) can be physically justified. Restricting ourselves to the initial states (15) will mean that for now, our computations of effective central charge will only be valid for some specific and simple types of quantum quenches, and generalizations to other scenarios are left for future projects.

It can be observed in [11], that given an initial state (15), one can compute correlation functions of primary fields, and at long times these relax to their thermal expected values, with an effective temperature $T_{\text{eff}} = 1/4\tau_0$. The relation between the extrapolation length and the effective temperature can be easily seen from computing the energy of the initial state, as

$$\frac{\langle\Psi_t|H|\Psi_t\rangle}{\langle\Psi_0|\Psi_0\rangle} = \frac{\pi c L}{24(2\tau_0)^2},$$

and comparing it the thermal ground state energy.

It is evident that for this CFT quench set up, the system thermalizes at late times, and the GGE reduces to the thermal ensemble. This is a consequence of choosing the initial state to be of the form (15). It was shown in [13,14] that more general states lead to a nontrivial GGE and richer dynamics at late times. The state (15), despite its simplicity, is particularly interesting since it was argued in [11] that it correctly describes a massive theory-to-CFT quench, when the pre-quench mass is very large, so there are some physical limits where it is relevant. This limit can be recovered explicitly in exactly solvable quantum quenches, such as the free bosonic field theory [11], and the scaling limit of the transverse field quantum Ising chain [15], where a Calabrese-Cardy type quench arises naturally in the limit $m/m_0 \to 0$, where $m$ is the post-quench mass.

Given the physical relevance and simplicity of the initial state (15), this is the only type of CFT quench we will consider in this paper. This will limit the range of quenches of massive theories we will be able to study later in this paper. We will later define an effective central charge for quenches of massive theories by comparing to this CFT result. This means that our definition of effective central charge of a massive theory will only be justified for quenches which reduce to the set up (15) in the appropriate CFT limit (when the post-quench massive parameter is very small). We will argue that this is indeed the case for some simple but physically relevant quench set ups of massive theories, such that our proposal has reasonable applicability. The study of more general and richer initial states is beyond the scope of this introductory paper, but it would be a very interesting problem to examine in the future.

The return amplitude can be computed for a CFT quench from initial state (15) by evaluating the partition function of the theory on the Euclidean strip. Such a partition function has been found in [16]. In the thermodynamic limit, $L \to \infty$, for some finite lengths $\tau_0$, $\tau$, the

strip partition function is given by

$$\frac{\langle \Psi_0^*|e^{-\tau H}|\Psi_0^*\rangle}{\langle \Psi_0^*|e^{-2\tau_0 H}|\Psi_0^*\rangle} = \exp\left(-\frac{\pi c L}{48\tau_0} + \frac{\pi c L}{24\tau}\right). \tag{16}$$

The return amplitude at real times is obtained from (16) by analytically continuing $\tau \to 2\tau_0 + it$,

$$\mathscr{F}(t) = \exp\left[\operatorname{Re}\left(-\frac{\pi c L}{48\tau_0} + \frac{\pi c L}{24(2\tau_0 + it)}\right)\right]. \tag{17}$$

The terms on the right hand side of (17) can be identified with the terms of the free energy (14), as

$$f_s = \frac{\pi c}{24(4\tau_0)}, \qquad f_C(\tau - 2\tau_0) = -\frac{\pi c}{24\tau}, \tag{18}$$

and $f_b = 0$.

The equations, (18) can be easily inverted to get two expressions for the central charge, continuing $\tau \to 2\tau_0 + it$,

$$c = \frac{24}{\pi T_{\text{eff}}} f_s = \operatorname{Re}\left[-\frac{12(2it + 1/T_{\text{eff}})}{\pi} f_C(it)\right], \tag{19}$$

which will be the basis of our definition of effective central charge for general quenches into field theories that are not conformal.

In general models, the effective central charge will be time dependent, and the two expressions in (19) will not be equivalent at all times. The guiding principles we need to follow to define an appropiate effective central charge from the generalized version of the expression (19) are:

1. The effective central charge must reduce to a constant value described by (19) when we take the CFT limit (vanishing post-quench mass).

2. The effective central charge should interpolate between the known equilibrium values at $t = 0$ and $t \to \infty$.

One natural definition of time-dependent effective central charge in an integrable field theory one might consider, consists on interpreting the difference between the two expressions in (19) to yield $\Delta c'_{\text{eff}}(t) = c_{\text{eff}}(t) - c_{\text{eff}}(0)$, such that we define

$$\Delta c'_{\text{eff}}(t) \equiv \operatorname{Re}\left[\frac{24}{\pi T_{\text{eff}}} f_s + \frac{12(2it + 1/T_{\text{eff}})}{\pi} f_C(it)\right]. \tag{20}$$

We will see, however, that this expression (20) is problematic, when we evaluate it for a mass quench of a free boson, in that at late times it permanently oscillates, instead of converging to the expected equilibrium value. Despite this, it is easy to see that while the value $c'_{\text{eff}}(t)$ keeps permanently oscillating, its time-averaged value, around which it oscillates, converges to the expected equilibrium value. Therefore, it seems that an appropiate definition for effective central charge in a non-conformal theory after a quench is instead given by time-averaged expression

$$\Delta c_{\text{eff}}(t) = \langle \Delta c'_{\text{eff}}(t)\rangle_t \equiv \frac{1}{t}\int_0^t dt'\, \Delta c'_{\text{eff}}(t'). \tag{21}$$

We remind the reader that the proposed effective central charge (21) was derived using the fact that the CFT at late times thermalizes, and there is only one parameter, $T_{\text{eff}}$, which describes the late-time dynamics. It is then expected to be applicable in general for quenches of massive theories, only if these quenches have some "CFT limit", (typically considering the pre-quench mass to be much larger than the post-quench mass), where the late time dynamics become approximately thermal. We will see in the next sections such a limit exists for the quench set ups we will consider, which are "mass quenches" where one changes the value of the mass scale at $t = 0$.

When the post-quench mass is not zero, and the pre-quench mass is finite, the system does not generally thermalize. The late time limit, $c_{\text{eff}}(t \to \infty)$ of our proposal (21) can then be taken as a definition of the effective central charge that corresponds to a given GGE state. As we increase the value of the pre-quench mass, we will observe that the final state goes to a thermal ensemble, and that $c_{\text{eff}}(t \to \infty)$ should then reproduce the thermal value, $c_{\text{thermal}}(1/T_{\text{eff}})$, as defined in Section 2.

The rest of this paper is devoted to examining the implications of the formula (21) for quenches in different integrable field theories. We will see that this proposal satisfies the several properties that are expected from a sensible definition of effective central charge. We will point out, however, that there is some ambiguity in this definition for IFT's, since for general quenches into a massive theory, the system does not thermalize, so the concept of an effective temperature, $T_{\text{eff}}$ has to be carefully defined.

## 5 Return amplitude in IFT quenches

We now consider quantum quenches where the post-quench Hamiltonian, $H$ describes an IFT with mass $m$. We can still compute the return amplitude in this case by considering the partition function of the theory on a strip, and analytically continuing to real times. The computation of the partition function can be done analytically for initial states that correspond to integrable boundary conditions [17] (for brevity, from now on we will use the term "integrable boundary states" to refer to such initial states).

Integrable boundary states have been shown to be of the form [19],

$$|\Psi_0\rangle = \exp\left(\int_0^\infty \frac{d\theta}{2\pi} K(\theta) A^\dagger(-\theta) A^\dagger(\theta)\right)|0\rangle, \tag{22}$$

where $|0\rangle$ is the ground state of $H$, and $A^\dagger(\theta)$ are the corresponding particle creation operators, and $K(\theta)$ is a function which satisfies the so-called "cross-unitarity condition":

$$K(\theta) = S(2\theta)K(-\theta).$$

The excitations in the initial state (22) consist of pairs of particles with equal energies, and opposite momenta. When one rotates the theory to the Euclidean strip, one can exchange the role of the (imaginary) time and spatial dimensions, which implies that the function $K(\theta)$ is related to the boundary reflection matrix, $R(\theta)$ by $K(\theta) = R(i\pi/2 - \theta)$.

The partition function in the Euclidean strip can be computed by employing a version of the thermodynamic Bethe ansatz with open, instead of periodic boundary conditions [17]. We refer to this formalism as the Boundary TBA (BTBA). We will not derive the results of the BTBA, but simply cite the needed formulas, which can be found in [17, 18]. The free energy function, $f_C(\tau)$, can be computed in the BTBA formalism by solving the set of integral equations [18]

$$f_C(\tau) = \mp \frac{m}{4\pi} \int_{-\infty}^\infty d\theta \cosh\theta H(\theta, \tau),$$

where

$$H(\theta, \tau) = \log\left[1 \pm |K(\theta)|^2 e^{-\varepsilon(\theta,\tau)}\right],$$

and $\varepsilon(\theta, \tau)$ is the solution of

$$\varepsilon(\theta, \tau) = 2m\tau\cosh\theta \mp \int_{-\infty}^{\infty} d\theta\, \varphi(\theta - \theta')H(\theta', \tau).$$

The $\pm$ signs are chosen to match the sign of $-S(0)$. We can normalize the initial state such that $2f_s = -f_C(0)$.

We can now write down the formula for the effective central charge for a quench in an IFT from an integrable boundary state by analytically continuing to real time and using (21),

$$c_{\text{eff}}(t) = \frac{1}{t}\int_0^t dt'\, c'_{\text{eff}}(t'),$$

with

$$c'_{\text{eff}}(t) = c_{\text{eff}}(0) + \text{Re}\left[\frac{24f_s}{\pi T_{\text{eff}}} \mp \frac{3m(2it + 1/T_{\text{eff}})}{\pi^2}\int_{-\infty}^{\infty} d\theta\, \cosh\theta H(\theta, it)\right]. \tag{23}$$

In the following section we will analyze the formula (23) for a mass quench of a free bosonic theory, where as expected, we see that the effective central charge interpolates between $c_{\text{eff}}(t = 0) = 0$, and $c_{\text{eff}}(t \to \infty) \approx 1$ for very energetic quenches. We will also address the issue of defining an effective temperature $T_{\text{eff}}$ in quenches into a massive theory.

# 6 Mass quench of a free boson

We now consider a quantum quench of a free massive boson, where for $t < 0$, the particle mass is $m_0$, and at $t = 0$ we suddenly switch the mass to $m$. In this simple quench, it is possible to find an exact expression for the initial state, which is of the form (22), by performing a simple Bogoliubov transformation between the pre-quench and post-quench particle creation operators.

We denote by $A_0^\dagger(p)$, $A_0(p)$, and $A^\dagger(p)$, $A(p)$ the pre- and post-quench creation and annihilation operators, respectively, which create or destroy a particle with momentum $p$. We consider the initial state, $|\Psi_0\rangle$ to be the ground state of the pre-quench Hamiltonian, defined as

$$A_0(p)|\Psi_0\rangle = 0. \tag{24}$$

The relation between pre-and post-quench operators is obtained by demanding that the bosonic field and its canonical momentum conjugate field be continuous at the $t = 0$ boundary, from which it can be found,

$$A_0(p) = c_p A(p) - d_p A^\dagger(-p), \quad A_0^\dagger(p) = c_p A^\dagger(p) - d_p A(-p),$$

where

$$c_p = \frac{1}{2}\left(\sqrt{\frac{E_p}{E_{0p}}} + \sqrt{\frac{E_{0p}}{E_p}}\right), \qquad d_p = \frac{1}{2}\left(\sqrt{\frac{E_p}{E_{0p}}} - \sqrt{\frac{E_{0p}}{E_p}}\right),$$

with $E_p = \sqrt{m^2 + p^2}$ and $E_{0p} = \sqrt{m_0^2 + p^2}$. Solving (24) in the basis of post-quench operators, yields an initial state given by (22) with

$$K(\theta) = \frac{\sinh(\theta - \xi)}{\sinh(\theta + \xi)}, \tag{25}$$

where $\xi$ is the pre-quench rapidity defined by $m_0 \sinh\xi = m \sinh\theta$.

It is easy to see that the concept of effective temperature, $T_{\text{eff}}$ is generally not as simple as in the CFT case. In a CFT quench, the effecive temperature is related to the extrapolation length, which regularizes the idealized Dirichlet boundary state (an approach which was explored for massive IFT's in [20]). As was shown in [21], we can attempt to read-off an extrapolation length from the solution (25), by trying to rewrite the initial state (22) in the form (15). The only way one can write the initial state as (15), however, is using a momentum-dependent extrapolation length, $\tau_0(\theta)$, which solves

$$K(\theta) = e^{-2m\cosh\theta\,\tau_0(\theta)} K_{\text{Dirichlet}}(\theta), \tag{26}$$

where for a free boson, the Dirichlet boundary conditions are simply $K_{\text{Dirichlet}}(\theta) = 1$. Using (25), we find

$$\tau_0(\theta) = -\frac{1}{2m\cosh\theta} \log\left[\frac{\sinh(\theta - \xi)}{\sinh(\theta + \xi)}\right]. \tag{27}$$

As was explained in [22], long times after the quench, the system behaves as if every momentum mode thermalizes with a different temperature, given by $T_{\text{eff}}(\theta) = 1/4\tau_0(\theta)$.

Since the effective temperature depends on the momentum modes, there is some ambiguity regarding what single number one should use as $T_{\text{eff}}$ in the definition of effective central charge (23). We will discuss here several definitions that were proposed in [22] that could be used in (23). For most of this paper, however, we will focus only on very energetic quenches, where $m_0/m \to \infty$, where the momentum-dependence of $\tau_0$ dissapears.

As is expected from the analysis of CFT quenches, in the limit $m/m_0 \to \infty$, the expression (27) becomes the constant

$$\tau_0(\theta) \to \frac{1}{m_0}, \tag{28}$$

such that we can use simply $T_{\text{eff}} = m_0/4$ in (23).

For general values of $m_0$, $m$, two different proposals for effective temperature were given in [22]. The first reasonable alternative is to use the value $T_{\text{eff}} = 1/4\tau_0(0)$, since the effective temperature of the zero momentum mode gives a good description of the long distance behaviour of any correlation function. Coincidentally, from (27), one finds $\tau_0(0) = 1/m_0$, which agrees with (28). It is easy to see that in general, the effective temperature for the zero-momentum mode, $\tau_0(0)$, serves as a bound for the other momentum modes, such that $\tau_0(\theta) \leq \tau_0(0)$, for all $\theta$.

A second definition of effective temperature proposed in [22] is chosen such that the thermal expectation value of a given local operator, $\mathcal{O}$, agrees with the long time expectation value of the same operator after the quench. Explicitly, one defines the temperature $T_{\text{eff}}^{\mathcal{O}}$ such that

$$\lim_{t \to \infty} \frac{\langle \Psi_t | \mathcal{O} | \Psi_t \rangle}{\langle \Psi_0 | \Psi_0 \rangle} = \langle \mathcal{O} \rangle_{T_{\text{eff}}^{\mathcal{O}}},$$

where the right hand side denotes the thermal expectation value, computed with temperature $T_{\text{eff}}^{\mathcal{O}}$. Such a computation involves an average over all the momentum modes, so the effective

temperature obtained this way is called "average effective temperature" in [22]. One disadvantage of this definition of effective temperature is that it is operator-dependent. For the operator $\mathcal{O} = \phi^2$ evaluated in [21] (where $\phi$ is the free bosonic field), it was shown that $T_{\text{eff}}^{\phi^2} \to m_0/4$ for $m_0/m \to \infty$, so all of these reasonable definitions of effective temperature converge.

It is beyond the scope of this paper to determine which is the best definition of $T_{\text{eff}}$ to include in our definition of effective central charge for general quenches. From now on, we will only be concerned with highly energetic quenches, where we will consider simply $T_{\text{eff}} = m_0/4$.

Let us now briefly discuss what is the behavior we expect from a reasonable definition of time-dependent effective central charge in the mass quench of a free boson. Since the initial state is the ground state of a massive theory, we expect $c_{\text{eff}}(t = 0) = 0$. At infinite times, the system locally equilibrates, and is described by the effective temperature $T_{\text{eff}}$, so we expect $c_{\text{eff}}(t \to \infty) = c_{\text{thermal}}(1/T_{\text{eff}}) \approx 1$ (this equivalence is only valid for $m_0/m \to \infty$).

For $m_0 \gg m$ we can write

$$K(\theta) \approx e^{-2\frac{m}{m_0}\cosh\theta},$$

such that from the BTBA we have

$$f_C(\tau) = \frac{m}{4\pi} \int_{-\infty}^{\infty} d\theta \cosh\theta \log\left[1 - e^{-\left(4\frac{m}{m_0} + 2m\tau\right)\cosh\theta}\right]. \tag{29}$$

We can then compute the function

$$
\begin{aligned}
c_{\text{eff}}'(t) &= -\frac{48}{\pi m_0} f_C(0) + \frac{12}{\pi} \text{Re}\left[(2\mathrm{i}t + 4/m_0) f_C(\mathrm{i}t)\right] \\
&= c_{\text{thermal}}\left(\frac{4}{m_0}\right) - \frac{6}{\pi^2} \text{Re}\left\{m(2\mathrm{i}t + 4/m_0) \sum_{n=1}^{\infty} \frac{1}{n} K_1\left[nm\left(\frac{4}{m_0} + \mathrm{i}2t\right)\right]\right\}, \tag{30}
\end{aligned}
$$

where we have set $c_{\text{eff}}(t = 0) = 0$. We can now show that while the function $c_{\text{eff}}'(t)$ continues oscillating permanently at long times, the time-averaged function $c_{\text{eff}}(t)$ converges to the thermal value, $c_{\text{thermal}}(4/m_0)$.

The function $c_{\text{eff}}'(t)$ can be studied at late times by using the asymptotic expressions for the Bessel functions (11), so we write

$$c_{\text{eff}}'(t) \approx c_{\text{thermal}}\left(\frac{4}{m_0}\right) - \frac{6}{\pi^2} m\sqrt{t} \sum_{n=1}^{\infty} \frac{1}{n^{3/2}} e^{-n4m/m_0}\left[\cos(n2mt) - \sin(n2mt)\right]. \tag{31}$$

The expression (31) at late times consists of a sum of permanently oscillating terms, which grow with an overall factor of $\sqrt{t}$. If we consider instead the time-averaged function, $c_{\text{eff}}(t)$, one can simply see that the time average of the late-time cosine and sine terms is zero, such that

$$c_{\text{eff}}(t \to \infty) = c_{\text{thermal}}\left(\frac{4}{m_0}\right),$$

as is expected.

We plot both functions, $c_{\text{eff}}'(t)$ and $c_{\text{eff}}(t)$ in Figure 2, where we can explicitly observe the behavior we describe; $c_{\text{eff}}'(t)$ oscillates and grows as $\sqrt{t}$, and $c_{\text{eff}}(t)$ seems to converge to $c_{\text{thermal}}(4/m_0)$ at late times.

In the case where the pre- and post-quench mass difference is not very large, the system is not expected to thermalize at long times, but to be described by some GGE state. It is easy

(a)

(b)

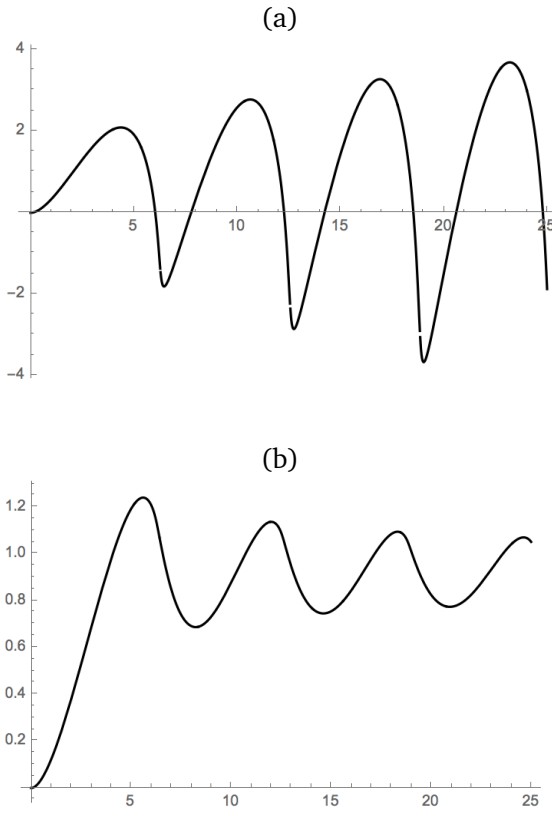

Figure 2: (a) Plot of $c'_{\text{eff}}(t)$ as a function of $t$. (b) Plot of the effective central charge, $c_{\text{eff}}(t)$, as a function of $t$. In both plots we have chosen the constants $4m/m_0 = 0.1$, and $2m = 1$.

to see by similar arguments to those above that in these cases the effective central charge also approaches some steady value at late times, given by $c_{\text{eff}}(t \to \infty) = -\frac{48 T_{\text{eff}}}{\pi} f_C(0)$, for some chosen quantity $T_{\text{eff}}$ (which as we argued could be, for example, the effective temperature of the zero-momentum mode, $\tau_0(0)$). We propose this expression can be considered as a definition of the effective central charge corresponding to some GGE state, $c_{\text{GGE}}[K(\theta)]$, which is a function of all the effective temperatures of all the momentum modes. The effective central charge corresponding to a given GGE with some effective temperature $\tau_0(0)$ is always smaller or equal to the analogous thermal value of central charge corresponding to the same temperature, $c_{\text{thermal}}(1/\tau_0(0))$. This is because of the bound $\tau_0(\theta) \leq \tau_0(0)$, which means all the non-zero momentum modes have a smaller or equal effective temperature in the GGE. When one considers the large quench limit, the GGE and thermal central charges converge, or $c_{\text{GGE}}[K(\theta)] \to c_{\text{thermal}}(1/\tau_0(0))$, for $m_0/m \to \infty$.

It is interesting to notice that as $c_{\text{eff}}(t)$ approaches its late-times asymptotic value, it oscillates and can reach values greater than $c_{UV} = 1$ during its finite time evolution. This phenomenon cannot be seen in equilibrium thermodynamics, since there is no real value of temperature for which $c_{\text{thermal}}(1/T) > 1$.

## 7 Effective quench from Ising to tricritical Ising CFT

In this section we study a quantum quench where the effective central charge flows from $c_{\text{eff}}(0) = 1/2$ to $c_{\text{eff}}(\infty) \approx 7/10$. These asymptotic values of central charge correspond to the first two unitary minimal CFT models, namely the Ising model (IM) and the tricritical Ising

model (TIM).

## 7.1 Massless flow from Tricritical Ising to Ising model at thermal equilibrium

The unitary minimal models, $\mathscr{M}_p$, labeled by an integer, $p \geq 3$, have a central charge given by [24]

$$c_p = 1 - \frac{6}{p(p+1)}, \tag{32}$$

with IM and TIM corresponding to the models $\mathscr{M}_3$ and $\mathscr{M}_4$, respectively. The primary operators of these models, $\Phi_{r,s}$, labeled by integers, $r,s$ have conformal weight

$$\Delta_{r,s} = \frac{((p+1)r - ps)^2 - 1}{4p(p+1)}.$$

It is known that if one perturbs the action of the unitary minimal model, $\mathscr{M}_p$ with the operator $\Phi_{1,3}$ (with a positive coupling constant) this will result in an integrable theory of massless particles describing the RG flow between $\mathscr{M}_p$ and $\mathscr{M}_{p-1}$ [6]. We are therefore interested in the integrable model with Hamiltonian

$$H = H_{\mathscr{M}_4} + \lambda \int dx \Phi_{1,3}. \tag{33}$$

This deformation introduces a mass scale $M \sim \lambda^{2\Delta_{1,3}-2}$.

The spectrum of the Hamiltonian (33) consists of massless fermionic particles, and bosons with mass $M$. The massive bosonic particles are highly unstable, so the thermodynamics of this model is dominated by the stable massless particles.

In the IR regime (for energies smaller than $M$), the theory (33) can be described as a free massless fermion deformed by an irrelevant operator, with action

$$
\begin{aligned}
S \;=\; & \frac{1}{2\pi} \int \left( \psi \partial_{\bar{z}} \psi + \bar{\psi} \partial_z \bar{\psi} \right) dz d\bar{z} \\
& - \frac{1}{\pi^2 M^2} \int (\psi \partial_z \psi)(\bar{\psi} \partial_{\bar{z}} \bar{\psi}) dz d\bar{z} + (\text{higher dimensional operators}),
\end{aligned} \tag{34}
$$

where the massless fermions have been divided into left- and right- chirality components, $\psi$ and $\bar{\psi}$, respectively.

The S-matrix for the massless fermions of the model (33) has been determined in [6]. The massless particles can be divided into right movers and left movers, which all move at the speed of light. The energy and momentum of left and right movers can be parametrized using a rapidity variable, as

$$E(\theta) \;=\; -p(\theta) = \frac{1}{2} M e^{-\theta}, \quad \text{for left movers,}$$

$$E(\theta) \;=\; p(\theta) = \frac{1}{2} M e^{\theta}, \quad \text{for right movers.}$$

Left and right movers can be created with the operators $A_L^\dagger(\theta)$ and $A_R^\dagger(\theta)$, respectively, which satisfy the algebra

$$
\begin{aligned}
A_L^\dagger(\theta_1) A_L^\dagger(\theta_2) &= -A_L^\dagger(\theta_2) A_L^\dagger(\theta_1), \\
A_R^\dagger(\theta_1) A_R^\dagger(\theta_2) &= -A_R^\dagger(\theta_2) A_R^\dagger(\theta_1), \\
A_R^\dagger(\theta_1) A_L^\dagger(\theta_2) &= S(\theta_1 - \theta_2) A_L^\dagger(\theta_2) A_R^\dagger(\theta_1),
\end{aligned} \tag{35}
$$

with the S-matrix

$$S(\theta) = -\tanh\left(\frac{\theta}{2} - \frac{i\pi}{4}\right). \tag{36}$$

Thermodynamical quantities at equilibrium in the model (33) can be computed by fixing two pseudo-energy functions, $\varepsilon_1(\theta)$ and $\varepsilon_2(\theta)$, corresponding to the right and left moving fermions, respectively. These can be computed by solving the massless version of the TBA integral equation, which yields [6]

$$\begin{aligned}
\varepsilon_1(\theta) &= \frac{1}{2}MRe^{\theta} - \int \frac{d\theta'}{2\pi}\varphi(\theta - \theta')\log\left(1 + e^{-\varepsilon_2(\theta')}\right), \\
\varepsilon_2(\theta) &= \frac{1}{2}MRe^{-\theta} - \int \frac{d\theta'}{2\pi}\varphi(\theta - \theta')\log\left(1 + e^{-\varepsilon_1(\theta')}\right),
\end{aligned} \tag{37}$$

where we use

$$\varphi(\theta) = -i\frac{d\log S(\theta)}{d\theta} = \frac{1}{\cosh(\theta)}.$$

The system of equations (37) is simplified by noticing the symmetry $\varepsilon_1(\theta) = \varepsilon_2(-\theta)$, which allows one to write these as a single integral equation in terms of $\varepsilon = \varepsilon_1$,

$$\varepsilon(\theta) = \frac{1}{2}MRe^{\theta} - \int \frac{d\theta'}{2\pi}\varphi(\theta + \theta')\log\left(1 + e^{-\varepsilon(\theta')}\right). \tag{38}$$

The thermal effective central charge is given by

$$c_{\text{thermal}}(R) = \frac{3MR}{2\pi^2}\int_{-\infty}^{\infty}d\theta\, e^{\theta}\log\left(1 + e^{-\varepsilon(\theta)}\right). \tag{39}$$

It is expected that the high-energy dynamics of the theory (33) is dominated by the TIM, with $c = 7/10$, and at low energies, the dynamics should be effectively that of the IM with $c = 1/2$. This desired behavior is indeed observed from (39), where it can be seen that $c_{\text{thermal}}(0) = 7/10$ and $c_{\text{thermal}}(\infty) = 1/2$ [6].

The high temperature limit of (39) can be extracted by observing that for small $MR$, the function $L(\theta) = \log\left(1 + e^{-\varepsilon(\theta)}\right)$ has the shape of a plateau with

$$L(\theta) = \begin{cases} 0, & \text{for } \theta \gg \log\left(\frac{2}{MR}\right), \\[2mm] \log\left(1 + e^{-\varepsilon_0}\right), & \text{for } -\log\left(\frac{2}{MR}\right) \ll \theta \ll \log\left(\frac{2}{MR}\right), \\[2mm] \log(2), & \text{for } \theta \ll -\log\left(\frac{2}{MR}\right), \end{cases} \tag{40}$$

where $\varepsilon_0$ is the solution of the transcendental equation

$$\varepsilon_0 = -\frac{1}{2}\log\left(1 + e^{-\varepsilon_0}\right).$$

The integral in (39) can be shown to localize and receive contributions only from the edges of the plateau (40), $\theta \approx \pm\log\left(\frac{2}{MR}\right)$, which yields the desired result $c_{\text{thermal}}(0) = 7/10$. The details of this computation can be found in [6].

In the $MR \to \infty$ limit, we can use the approximation $\varepsilon(\theta) = \frac{1}{2}MRe^{\theta}$, such that

$$\lim_{R \to \infty} c_{\text{thermal}}(R) = \frac{3MR}{2\pi^2}\int_{-\infty}^{\infty}d\theta\, e^{\theta}\log\left(1 + e^{-\frac{1}{2}MRe^{\theta}}\right) = \frac{1}{2}.$$

## 7.2 Massless flow out of equilibrium

We now consider a quantum quench of the model (33) where at $t = 0$ we suddenly switch the mass parameter from $M_0$ to $M$. We consider the initial state to be the ground state of the pre-quench Hamiltonian. The initial effective central charge, $c_{\text{eff}}(0)$ therefore corresponds to the zero-temperature value of (39), *i.e.* $c_{\text{eff}}(0) = 1/2$.

We will only be interested in the limit of a very energetic quench, where $M_0 \gg M$, such that at infinite times, the system is described by a large effective temperature, $T_{\text{eff}} = M_0/4 \gg M$. We then expect $\lim_{t \to \infty} c_{\text{eff}}(t) \approx 7/10$.

In the limit of the highly energetic quench, similarly to the quenches of CFT's, we assume that the initial state can be properly described by applying an extrapolation length to the Dirichlet boundary state:

$$|\Psi_0\rangle \approx e^{-H\tau_0}|D\rangle.$$

This means that physical observables can be computed by approximating the initial state to be given by (22), with the function $K(\theta)$ dominated only by the extrapolation length,

$$|K(\theta)| \approx e^{-\frac{M}{M_0}e^\theta}, \tag{41}$$

where we have taken $\tau_0 = 1/M_0$.

The initial state given by (41) can be used in a massless version of the BTBA, such that the boundary free energy is given by

$$f_C(\tau) = -\frac{M}{4\pi} \int_{-\infty}^{\infty} d\theta \frac{1}{2} e^\theta \log\left[1 + e^{-\varepsilon(\theta,\tau)-2\frac{M}{M_0}e^\theta}\right],$$

with

$$\varepsilon(\theta,\tau) = M\tau e^\theta - \int_{-\infty}^{\infty} \varphi(\theta-\theta')\log\left[1 + e^{-\varepsilon(\theta,\tau)-2\frac{M}{M_0}e^\theta}\right],$$

and $2f_s = -f_C(0)$. We can then define the function

$$c'_{\text{eff}}(t) = \frac{1}{2} + c_{\text{thermal}}(4/M_0) - \text{Re}\left\{\frac{3M(2it+4/M_0)}{\pi^2} \int_{-\infty}^{\infty} d\theta \frac{1}{2} e^\theta \log\left[1 + e^{-\varepsilon(\theta,it)-2\frac{M}{M_0}e^\theta}\right]\right\}, \tag{42}$$

and we define the effective central charge as the time average of (42).

It is easy to show that $c'_{\text{eff}}(t)$ as given in (42) flows from $1/2$ at $t = 0$, to $7/10$ at $t \to \infty$, for $M \ll M_0$. At $t = 0$, the third term in the right-hand side of (42) reduces to the thermal value $-c_{\text{thermal}}(4/M_0)$, given in (39), cancelling with the second term in the right hand side of (42). At very large times, we can approximate

$$\varepsilon(\theta,it) \to M\,it\,e^\theta,$$

so that

$$\begin{aligned}
\lim_{t\to\infty} c'_{\text{eff}}(t) &= \frac{1}{2} + c_{\text{thermal}}(4/M_0) \\
&\quad - \text{Re}\left\{\frac{3M(2it+4/M_0)}{\pi^2} \int_{-\infty}^{\infty} d\theta \frac{1}{2} e^\theta \log\left[1 + e^{-(2it+4/M_0)\frac{1}{2}Me^\theta}\right]\right\} \\
&= \frac{1}{2} + c_{\text{thermal}}(4/M_0) - \frac{1}{2}.
\end{aligned}$$

Large values of $M_0$ correspond to a large effective temperature, which means, as we discussed in the previous subsection, that $c_{\text{thermal}}(4/M_0) \approx 7/10$, as expected.

Since the function $c'_{\text{eff}}(t)$ in this case already approaches a constant value at late times, The time-averaged effective central charge, $c_{\text{eff}}(t)$ will converge to the same constant value at late times. Therefore we confirm that the function $c_{\text{eff}}(t)$ indeed interpolates between $1/2$ at $t = 0$ and $7/10$ at $t \to \infty$, as is expected.

It is interesting to notice that while it is relatively straightforward to design a quantum quench where the central charge flows from the IM value of $1/2$, to the TIM value of $7/10$, it seems to be impossible to design a quench where the central charge flows from TIM to IM. An initial state such that $c_{\text{eff}}(t = 0) = 7/10$, corresponds to considering the theory with Hamiltonian (33) at very high temperatures. If we want that at infinite times, $c_{\text{eff}}(t \to \infty) \approx 1/2$, we would need the late-time dynamics to resemble the low-temperature regime of (33), that is, the quantum quench must introduce a large negative amount of energy into the system.

Quantum quenches from a thermal initial state in a free theory have been studied in [22]. While it was shown that a quantum quench can indeed introduce negative energy into the system (the effective temperature at late times is lower than the temperature of the initial state, therefore called a "cold quench" in [22]), it was found that the amount of negative energy a quench can introduce is bounded. In a cold quantum quench of a free boson from a thermal initial state, the lowest value the effective temperature at late times can be is half of the initial temperature. If this result can be generalized to interacting theories, it would imply that if one starts at a very high initial temperature (such that $c_{\text{thermal}} \approx 7/10$ in our model), a cold quench cannot reduce the effective temperature down to a value low enough such that $c_{\text{eff}}(t \to \infty) = 1/2$.

We can then conclude that it is not difficult to design a quench where the dynamics effectively flow from IM to TIM. The reversed quench, flowing from TIM to IM, seems to be impossible with our methods. The results of this section can also be easily generalized to quenches between any two adjacent unitary minimal models, $\mathcal{M}_p$ and $\mathcal{M}_{p+1}$, by considering a theory with Hamiltonian

$$H = H_{\mathcal{M}_{p+1}} + \lambda \int dx \, \Phi_{1,3}. \tag{43}$$

The Hamiltonian (43) describes the massless RG flow between the minimal models $\mathcal{M}_p$ and $\mathcal{M}_{p+1}$. The TBA equations for the massless particles of (43) was found in [6], where it can be shown that at low temperatures, the effective central charge is that of $\mathcal{M}_p$, and at high temperatures, the effective central charge is that of $\mathcal{M}_{p+1}$. It is then easy to design a quantum quench where the time-dependent effective central charge flows from the value corresponding to $\mathcal{M}_p$ at $t = 0$, to a value approaching that of $\mathcal{M}_{p+1}$ at $t \to \infty$. This would be done by starting from the zero-temperature ground state of (43), and suddenly making a large change in the $\lambda$ parameter, inducing a large effective temperature.

## 8 Quench into the staircase model

In this section we will study quantum quenches into the staircase model, which was introduced in [7]. The staircase model can be understood as an analytic continuation of the sinh-Gordon model, with action

$$S = \int d^2x \left( \frac{1}{2} \partial_\mu \phi \partial^\mu \phi - \frac{m^2}{g^2} \cosh g\phi \right),$$

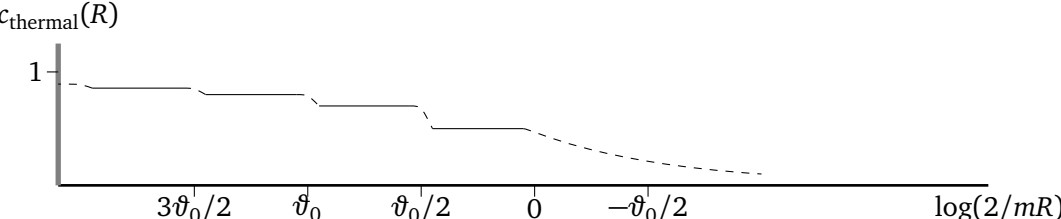

Figure 3: Schematic plot (note that this is not an actual numerical evaluation of the TBA equations) of the thermal central charge in the staircase model as a function of $\log(2/mR)$. For high temperatures, the central charge reaches plateaus given by the central charges of the unitary minimal models.

where $m$ is a mass scale and $g$ the coupling constant. This is an integrable model with a single species of massive particles, and the S-matrix is known to be

$$S(\theta) = \frac{\sinh\theta - i\sin\gamma}{\sinh\theta + i\sin\gamma},$$

where we define the constant $\gamma = \pi g^2/(8\pi + g^2)$. This S-matrix can be inserted into the TBA formalism, to compute the thermal effective central charge $c_{\text{thermal}}(R)$. It is not difficult to see that at low temperatures, $c_{\text{thermal}}(\infty) = 0$, while at high temperatures $c_{\text{thermal}}(0) = 1$, as is expected.

The staircase model is obtained from the sine-Gordon TBA equations by performing the analytic continuation $\gamma = \frac{\pi}{2} \pm i\vartheta_0$, where $\vartheta_0$ is real, and then letting $\vartheta_0$ tend to infinity. As we will see in the following subsection, the thermal effective central charge, $c_{\text{thermal}}(R)$ still interpolates between the values of 0 and 1 at low and high temperatures, however, as we increase the value of $\vartheta_0$, the effective charge develops a series of plateaus (or a "staircase") as a function of $mR$, as pictured in Figure 3. The values of the central charge at these plateaus are given by Eq. (32), corresponding to the central charges of all the unitary minimal models.

In the next subsection, we examine the TBA equations of the staircase model, and review how the staircase structure in $c_{\text{thermal}}(R)$ arises at large $\vartheta_0$. We then study the effective central charge after a quantum quench into the staircase model, for high energy quenches, with very large effective temperature. We find that at very early times after the quench, the effective central charge passes through a series of steps, however the values of the central charge at these steps are shifted away from the values in (32).

## 8.1 The staircase model at thermal equilibrium

The thermal effective central charge in the staircase model is given by Equations (8) and (9), using the kernel

$$\varphi(\theta) = -i\frac{d}{d\theta}\ln S(\theta) = \frac{1}{\cosh(\theta + \vartheta_0)} + \frac{1}{\cosh(\theta - \vartheta_0)}. \tag{44}$$

The steps in the central charge only appear in the $\vartheta_0 \to \infty$ limit, while simultaneously reducing the value of $mR$. In this limit, the kernel (44) acquires the shape of two individual localized lumps, centered around the values $\theta = \pm\vartheta_0$. Equation (8) then couples the pseudoenergy $\varepsilon(\theta)$ to $\varepsilon(\theta')$ with $\theta' \approx \theta \pm \vartheta_0$. At small values of $mR$, the function $L(\theta) = \log\left(1 + e^{-\varepsilon(\theta)}\right)$ acquires the shape of a plateau with non-zero values only in the interval $-\log(2/mR) \ll \theta \ll \log(2/mR)$, as pictured in Figure 4. It is then easy to see that as long as $\log(2/mR) \ll \vartheta_0/2$, the second term in equation (8) does not contribute, and the pseudo

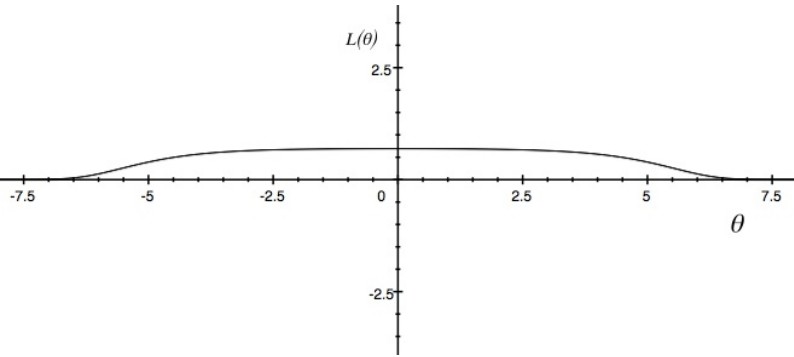

Figure 4: A plot of the function $L(\theta) = \log[1 + \exp(-mR\cosh\theta)]$, vs $\theta$, where we have chosen the small value $mR = .01$.

energy is given by $\varepsilon(\theta) = m\cosh\theta$, that is, the TBA equations are those of a free fermion. We then reach the first plateau in the staircase model by taking the high temperature limit, with very small $mR$, while keeping $\log(2/mR) \ll \vartheta_0/2$, where the central charge is given by the free fermion value, $c = 1/2$.

The second term in the right-hand side of (8) contributes to $\varepsilon(\theta)$ once the $L(\theta)$ plateau becomes wide enough to reach the values $\theta \approx \pm\vartheta_0$. The new contributions to the pseudo energy come from the two localized lumps of the $\varphi(\theta)$ kernel, which means, the pseudo energy can be divided in two contributions, $\varepsilon_1(\theta) = \varepsilon(\theta + \vartheta_0)$ and $\varepsilon_2(\theta) = \varepsilon(\theta - \vartheta_0)$. The TBA equations in this case turn out to be equivalent to the ones discussed in Section (7.1), with $\varepsilon_{1,2}(\theta)$ describing the left and right moving massless particles, which characterize the RG flow from TIM to IM. At high temperatures, in the region $\vartheta_0/2 \ll \log(2/mR) \ll \vartheta_0$, the central charge reaches the plateau at the value $c = 7/10$ corresponding to the TIM.

One can continue decreasing the value of $mR$, which has the effect of widening the non-zero region of the function $L(\theta)$. A change in behavior in the TBA equations occurs every time that $\log(2/mR)$ reaches the value of an integer multiple of $\vartheta_0/2$, where as was shown in [7], the pseudo energy can effectively split into a higher number of relocalized pseudo energies, which describe the massless flow between adjacent minimal models. In particular, if one focuses on the region where $(p-3)\vartheta_0/2 \ll \log(2/mR) \ll (p-2)\vartheta_0/2$, the effective central charge reaches the plateau at the value $c = 1 - 6/[p(p+1)]$, corresponding to the $\mathcal{M}_p$ minimal model.

## 8.2 The staircase model out of equilibrium

We now examine the staircase model after a quantum quench. In particular, we will consider the initial state to be the ground state of a free boson, of mass $m_0$. At time $t = 0$ we suddenly turn on the interaction term, with the value of coupling constant given by $\gamma = \frac{\pi}{2} + i\vartheta_0$, as well as simultaneously changing the particle mass to $m$. The reason we focus on this particular quench protocol is that the initial state corresponding to such a quench in the sinh-Gordon model has been studied in Ref. [21, 25], where it has been found that it is very well approximated by (22), with

$$K(\theta) = K_{\text{free}}(\theta)K_{\text{Dirichlet}}(\theta), \tag{45}$$

where $K_{\text{free}}(\theta)$ describes the mass quench of a free boson, and is given in Eq. (25); $K_{\text{Dirichlet}}(\theta)$ corresponds to the (non-normalizable) state given by considering Dirichlet boundary conditions, which can be found to be [26]

$$K_{\text{Dirichlet}}(\theta) = i\tanh(\theta/2)\frac{\cosh(\theta/2 - i\gamma/4)\sinh(\theta/2 + i(\gamma+1)/4)}{\sinh(\theta/2 + i\gamma/4)\cosh(\theta/2 - i(\gamma+1)/4)}.$$

We will assume that the initial state (45) is still valid when we analytically continue to complex values of the coupling constant. As we have done in previous sections, we will now focus only on very energetic quantum quenches, for which $m \ll m_0$. In this limit, the details of the initial state become less important, and we can approximate it with the simple extrapolation length,

$$|K(\theta)| \approx e^{-2\frac{m}{m_0}\cosh\theta},$$

for large values of $\theta$.

The initial state for the quench is the ground state of the Hamiltonian of a free massive boson. We therefore set $c_{\text{eff}}(t=0) = 0$. Using the formula (23), we can write the function

$$c'_{\text{eff}}(t) = \text{Re}\left[ \frac{12m}{\pi^2 m_0} \int_{-\infty}^{\infty} d\theta \cosh\theta H(\theta, 0) - \frac{3m(2it + 4/m_0)}{\pi^2} \int_{-\infty}^{\infty} d\theta \cosh\theta H(\theta, it) \right], \quad (46)$$

with

$$H(\theta, \tau) = \log\left[ 1 + e^{-4\frac{m}{m_0}\cosh\theta - \varepsilon(\theta, \tau)} \right],$$

and

$$\varepsilon(\theta, \tau) = 2m\tau\cosh\theta - \int_{-\infty}^{\infty} d\theta'\left[ \frac{1}{\cosh(\theta - \theta' + \vartheta_0)} + \frac{1}{\cosh(\theta - \theta' - \vartheta_0)} \right] H(\theta', \tau). \quad (47)$$

At long times, we have $\varepsilon(\theta, it) \approx 2mit\cosh\theta$. The integrand in the second term in the right-hand-side of (46) becomes highly oscillatory, and after time-averaging, its contribution to $c_{\text{eff}}(t)$ vanishes. The effective central charge at late times is then given by

$$\lim_{t\to\infty} c_{\text{eff}}(t) = \text{Re}\left[ \frac{12m}{\pi^2 m_0} \int_{-\infty}^{\infty} d\theta \cosh\theta H(\theta, 0) \right] = c_{\text{thermal}}\left( \frac{4}{m_0} \right).$$

We assume that the effective temperature, $T_{\text{eff}} = 4/m_0$ is very high, such that the value of $c_{\text{thermal}}(4/m_0)$ lies in one of the plateaus discussed in the previous subsections. To be more specific, we can select a value of $m_0$, such that $(p-3)\vartheta_0/2 \ll \log(m_0/2m) \ll (p-2)\vartheta_0/2$, for some integer, $p$, such that $c_{\text{thermal}}(4/m_0) \approx 1 - 6/[p(p+1)]$. This result arises from the fact that the function $H(\theta, 0)$ is the same as the $L(\theta)$ function discussed in the previous subsection. The function $H(\theta, 0)$ at very large $m_0$ acquires the shape of a plateau that is non-zero only in the region $-\log(m_0/2m) \ll \theta \ll \log(m_0/2m)$. The value of $c_{\text{eff}}(\infty)$ then depends on the integer number of times that $\vartheta_0$ fits in this interval.

We can now observe that the time evolution of the central charge described by (46) also exhibits a "staircase" structure at very short times, where the value of the effective central charge reaches a series of plateaus as a function of time. To see this staircase structure, we study the function $\text{Re}[H(\theta, it)]$ at very short times.

We first consider the function $\text{Re}[H(\theta, it)]$ in the regime $1 \ll \log(1/mt) \ll \vartheta_0/2 \ll \log(m_0/2m)$. In this regime, the function $\text{Re}[H(\theta, it)]$ aqcuires the shape shown in Figure 5. This function has a nearly constant value in the plateau given by $-\log(1/mt) \ll \theta \ll \log(1/mt)$. In the regions $-\log(m_0/2m) \ll \theta \ll -\log(1/mt)$ and $\log(1/mt) \ll \theta \ll \log(m_0/2m)$, the function of $\theta$ is very highly oscillatory. In the regions $\theta \ll -\log(m_0/2m)$ and $\log(m_0/2m) \ll \theta$, the function vanishes exponentially to zero.

We now argue that when we insert $H(\theta, it)$ into (47), and integrate over all $\theta$, the highly oscillatory, and exponentially vanishing regions in Figure 5 do not contribute to the function $\varepsilon(\theta, it)$. The only contributions to $\varepsilon(\theta, it)$ will come from the constant nonzero values of $\text{Re}[H(\theta, it)]$. As was discussed for the thermal central charge in the previous subsection, the

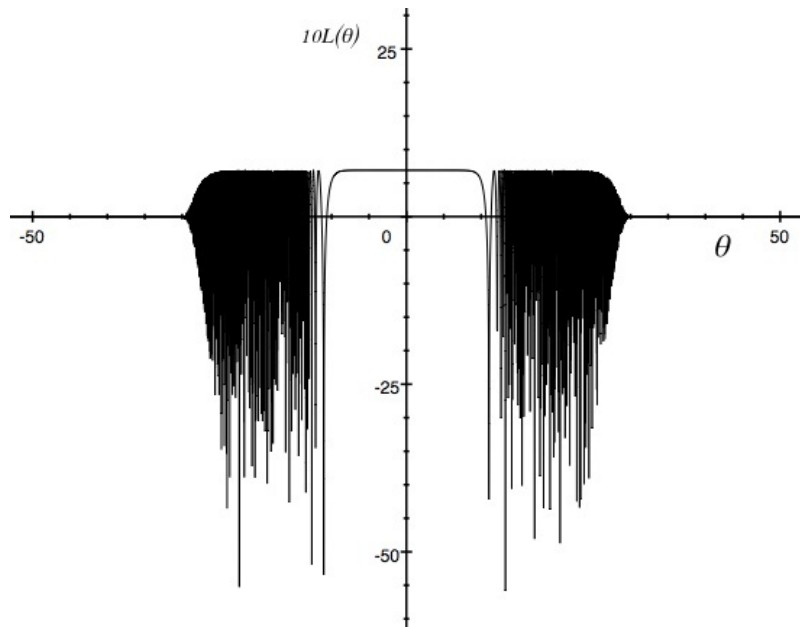

Figure 5: A plot of the function $10L(\theta) = 10\,\mathrm{Re}H(\theta, it) = 10\,\mathrm{Re}\log[1 + \exp((-4m/m_0 - 2mit)\cosh\theta)]$ (where the overall factor of 10 is only inserted for better visibility), vs $\theta$, where we have chosen the small values $4m/m_0 = 10^{-12}$ and $2mt = 10^{-4}$.

value of the effective central charge, $c_{\mathrm{eff}}(t)$ then simply depends on the integer number of times that the value of $\vartheta_0$ fits in the interval $(-\log(1/mt), \log(1/mt))$. The number of times that $\vartheta_0$ can fit in this interval increases as we reduce $mt$.

We now suppose that the pre-quench mass, $m_0$, is such that $(p-3)\vartheta_0/2 \ll \log(m_0/2m) \ll (p-2)\vartheta_0/2$. At time $t = 0$, the function $\mathrm{Re}[H(\theta, 0)]$ is dominated by the non-zero constant value in the plateau $-\log(m_0/2m) \ll \theta \ll \log(m_0/2m)$, and has no oscillatory behavior. At this point, the two terms in the right-hand side of (46) cancel each other, and $c_{\mathrm{eff}}(t=0) = 0$. The shape of $\mathrm{Re}[H(\theta, it)]$ remains unchanged until the time reaches the interval $(p-4)\vartheta_0/2 \ll \log(1/mt) \ll (p-3)\vartheta_0/2$, where now $\vartheta_0$ fits one less integer time in the constant, non-oscillatory region of $\mathrm{Re}[H(\theta, it)]$, given by $|\theta| \ll \log(1/mt)$. From (46), it is easy to see that within this time interval, the (non-time-averaged function) $c'_{\mathrm{eff}}(t)$ reaches the steady value

$$c'_{\mathrm{eff}}(t)|_{(p-3)\vartheta_0/2 \ll \log(1/mt) \ll (p-4)\vartheta_0/2} = -\frac{6}{p(p+1)} + \frac{6}{(p-1)p}.$$

The effective central charge will again remain at this constant value until the time reaches the next interval, $(p-5)\vartheta_0/2 \ll \log(1/mt) \ll (p-4)\vartheta_0/2$, where the central charge will increase to the next plateau. The function $c'_{\mathrm{eff}}(t)$ will change its value whenever the time enters a new interval $(p-k-3)\vartheta_0/2 \ll \log(1/mt) \ll (p-k-2)\vartheta_0/2$, for some integer $k$, such that $0 \le k \le p-3$. During such an interval, one finds

$$c'_{\mathrm{eff}}(t)|_{(p-k-3)\vartheta_0/2 \ll \log(1/mt) \ll (p-k-2)\vartheta_0/2} = c_p - c_{p-k}$$

$$= -\frac{6}{p(p+1)} + \frac{6}{(p-k)(p-k+1)}. \tag{48}$$

It is easy to see that when we time-average the function (48) to obtain the effective central charge, the same staircase structure is maintained for $c_{\mathrm{eff}}(t)$. This is because the staircase

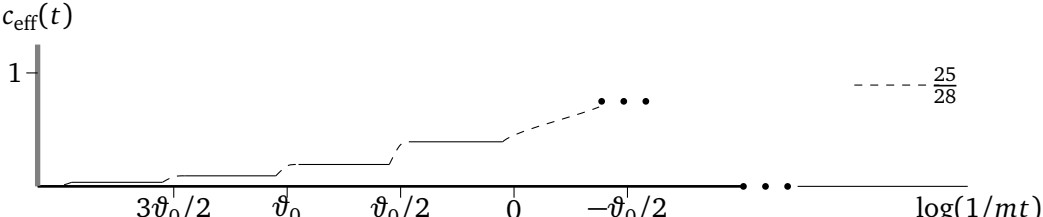

Figure 6: Schematic plot (note that this is not an actual numerical evaluation of Eq. (46)) of the time evolution of the effective central charge after a quantum quench into the staircase model. The plot corresponds to an initial mass $m_0$, such that $(p-3)\vartheta_0/2 \ll \log(m_0/2m) \ll (p-2)\vartheta_0/2$, with $p = 7$. At infinite times, the effective central charge reaches the stationary value, $c = 25/28$, corresponding to the $\mathscr{M}_7$ minimal model.

structure occurs in a logarithmic time scale. When we time-average in a linear time scale, the duration of each step of the staircase is much longer than the duration of all the previous steps, such that each steps dominates the time average. Explicitly, the function $c'_{\text{eff}}(t)$ stays in the $k$-th step of the staircase for the duration of the time interval $e^{-(p-k-2)\vartheta_0/2} \ll mt \ll e^{-(p-k-3)\vartheta_0/2}$, while the duration of all the previous steps combined is given by $0 \ll mt \ll e^{-(p-k-2)\vartheta_0/2}$. The ratio of the duration of the $k$-th step compared to the duration of all the previous steps, is then given by

$$\frac{\Delta t_{k-\text{th step}}}{\Delta t_{\text{previous steps}}} = \frac{e^{-(p-k-3)\vartheta_0/2} - e^{-(p-k-2)\vartheta_0/2}}{e^{-(p-k-2)\vartheta_0/2}},$$

which in the limit, $\vartheta_0 \to \infty$ becomes

$$\frac{\Delta t_{k-\text{th step}}}{\Delta t_{\text{previous steps}}} \to e^{\vartheta_0/2},$$

such that the duration of each step is exponentially larger than the combined duration of all previous steps. This implies that after time-averaging the function $c'_{\text{eff}}(t)$, it will preserve the same staircase structure, so the effective central charge satisfies

$$c_{\text{eff}}(t)|_{(p-k-3)\vartheta_0/2 \ll \log(1/mt) \ll (p-k-2)\vartheta_0/2} = -\frac{6}{p(p+1)} + \frac{6}{(p-k)(p-k+1)}. \tag{49}$$

The steps in the central charge given by (49) occur only at very short times after the quench. This behavior is analogous to how at thermal equilibrium, the steps in the central charge can only be observed at very high temperatures. In Figure 6, we present a schematic plot of the time evolution of the effective central charge for a given value of $p$.

## 9  Conclusions

We have proposed a definition for a time-dependent effective central charge that describes a massive field theory after a quantum quench. Quantum quenches from a pure initial state introduce an extensive amount of energy into the system, such that at long times the state can be described by some finite effective temperature(s). As is expected, the effective central charge at large times is higher or equal to that at $t = 0$, corresponding to a higher temperature state.

This general relation between the value of the effective central charge at $t = 0$ and $t \to \infty$ is tied to the fact that the thermal equilibrium effective central charge itself has constraints

related to the irreversibility of RG flow. In equilibrium, the effective central charge increases monotonically as a function of the energy scales that are probed. After a quantum quench starting from a pure state, one will always end up probing higher energy scales than in the pre-quench set up. Our definition of time-dependent effective central charge then captures this irreversibility property between the initial and final state, in a way that is analogous to how the thermal central charge captures the effects of RG irreversibility at thermal equilibrium. The function we have defined, however, can show oscillatory behavior at finite intermediate times, so it is difficult to give an RG interpretation of the meaning of this charge at finite times

The issue of irreversibility in quantum quenches has been previously addressed in terms of changes in entropy. Following the second law of thermodynamics, entropy in a closed system is always expected to increase. This can be explicitly confirmed in a quantum quench by computing the time evolution of different quantities that have been defined to measure entropy [27, 28]. Irreversibility has also been recently studied in the context of quantum manybody systems by examining the Loschmidt echo [29], which is a quantity closely related to the return-amplitude considered in this paper. The Loschmidt echo measures the overlap between the initial state of the system and a state that has been time-evolved, and then evolved backwards in time with slightly modified Hamiltonian. The Loschmidt echo is seen to generally decay exponentially with time in the examples of [29], indicating an irreversible process.

The effective central charge we defined provides a new complimentary characterization of irreversibility in quantum quenches, which may provide a connection with the concept of irreversibility of RG flow. In the future, it would be interesting to see if any explicit connections can be found between the effective central charge and quantities like the diagonal entropy, defined in [27]. This could provide a deeper understanding of relation between RG irreversibility and the second law of thermodynamics in quantum quenches

As a simple application of our proposed effective central charge, we considered a large mass quench of a free boson. As expected, the central charge interpolates between the IR and UV values of $c = 0$ and $c \approx 1$ at $t = 0$ and $t \to \infty$, respectively. Despite the irreversibility that characterizes the $t = 0$ and $t \to \infty$ values, at finite times, the effective central charge can oscillate, and does not necessarily increase monotonically.

For smaller mass quenches of a free boson, the system does not thermalize at late times, but locally relaxes into a GGE, where each momentum mode can have a different effective temperature. Our proposal at late times can then be used to define the concept of effective central charge corresponding to a GGE configuration. We argued this definition should be valid for quench set ups which have some limit where the post-quench dynamics are described by CFT (taking the post-quench mass to zero in our case), and the system is seen to thermalize at late times in this limit. This is indeed the case in the quenches we considered, where it is seen that for $m_0 \gg m$, the effective temperature approaches a constant value for every momentum mode. It would be interesting in the future to perform more detailed numerical studies, to understand precisely how the effective central charge depends on the infinite number of generalized temperature parameters involved in the GGE.

We defined the concept of a quantum quench which interpolates between two different CFT fixed points at $t = 0$ and $t \to \infty$. In particular, we examined a quench that interpolates between the Ising model (IM) with $c = 1/2$, and the tricritical Ising model (TIM) with $c = 7/10$. The quantum quench can only be done in the "IM to TIM" direction, and the reverse quench is impossible. Analogous conclusions can be made for quenches which interpolate between any two adjacent unitary minimal CFT's.

This result can be understood from the fact that the central charge can be interpreted as a measure of the number of local degrees of freedom of a CFT. The Hilbert space of the TIM is higher-dimensional than that of the IM. In this sense, a quantum quench starting from the IM ground state, and time-evolved into TIM dynamics is sensible, since the IM ground state "fits"

within the TIM Hilbert space, and can be readily expressed in terms of the TIM eigenstates. In the reverse direction, the ground state of the TIM has no interpretation in terms of the lower-dimensional IM Hilbert space.

We finally studied the evolution of the effective central charge in a quench into the staircase model. For highly energetic quenches, the effective central charge at $t \to \infty$ corresponds to that of a unitary minimal model, determined by the effective temperature. At very short times, the central charge evolves in an ascending "staircase" structure, where the values of central charge at each step can be computed in terms of the charges of minimal models. In this case, it seems that RG irreversibility may be reflected in the fact that the time evolution along this staircase can only be ascending in time, and never descending.

As we have stated, the interpretation of the effective central charge in terms of the irreversibility of RG flow after a quantum quench seems to be clear in terms the limiting values $c_{\text{eff}}(0)$ and $c_{\text{eff}}(t \to \infty)$. We are able to observe a simple physical principle that in a quench starting from a pure state, it is always true that $c_{\text{eff}}(\infty) \geq c_{\text{eff}}(0)$. This can clearly be interpreted as the fact that the quench always increases the energy scales at which the field theory is probed, such that the equilibrium state at late times should be described by a larger value of the c-function. The RG interpretation of $c_{\text{eff}}(t)$ at intermediate times is less obvious to us. Generally, the expectation is that this function should give us a measure of exactly how the system transforms, and what are the energy scales probed as the system evolves from a lower to higher value of effective central charge. At this point we can only speculate about the meaning of $c_{\text{eff}}(t)$ at finite times, and about what, if any, interpretation it may have in terms of RG flow. A more detailed numerical study of the effective central charge would be useful, and would hopefully provide some insight on configurations visited by the system at intermediate times, which may be characterized by the behavior of $c_{\text{eff}}(t)$.

It would be useful in the future to generalize our definition of effective central charge for quantum quenches that cannot be described by the Calabrese-Cardy initial states in the CFT limit (15). One logical extension of these initial states was proposed in [13], where one considers modifying the ideal conformally invariant initial state not only with with an extrapolation length, but with an infinite set of scales,

$$|\Psi_0\rangle = e^{-\sum_k \tau_0^k Q^k}|\Psi_0^*\rangle,. \tag{50}$$

where $Q^k$ are conserved quantities, and not only the Hamiltonian. It has been shown that a quantum quench from the state (50) leads to an effective GGE at late times, and not to a thermal state. To generalize our definition of effective central charge, we would need to compute the return amplitude of the CFT quench from the initial state (50), and examine how it depends on the central charge, $c$. We can then invert this function to define an effective central charge for the generalized quench. Such a generalized formula was not needed so far for this paper, since for our examples, we can see explicitly that in the CFT limit, $m/m_0 \to \infty$, the standard extrapolation length is dominant over all other scales in (50). There are however, physical quantum quenches beyond the scope of this paper, where the the CFT limit is not described simply by the extrapolation length. One simple example is the scaling limit of the Ising spin chain, [15] where our effective central charge is applicable only in the limit $m_0 \to \infty$, but not if $m/m_0 \to 0$, with finite $m$.

Finally, it would be interesting to see if the ideas we have introduced can be applied to higher dimensional field theories. For even-dimensional space-times, a quantity analogous to the thermal central charge was proposed in [30]. A proof that this quantity decreases monotonically along RG flow in four dimensions was acheived in [31]. It would be useful to see if our conclusions based on the effective central charge of 2d quantum quenches can be similarly applied for the analogous 4d function.

## Acknowledgements

I would like to thank Giuseppe Mussardo, Lorenzo Piroli, Dirk Schuricht and Eric Vernier, for many discussions and comments on this manuscript. This work is supported by the European Union's Horizon 2020 under the Marie Sklodowoska-Curie grant agreement 750092.

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
