# Peer review of "Time evolution of effective central charge and signatures of RG irreversibility after a quantum quench"

_SciPost Physics, doi:SciPost Phys. 4, 016 (2018)_

## Round 1 · Referee Report · Anonymous (Referee 1) · 2017-8-29

Strengths

1- A new theory is developed and the models analysed match well the expected physics 2- The discussion is quite clear 2- Interesting connection between RG irreversibility and quantum quench irreversibility

Weaknesses

1- The theory is, I think, of relatively limited applicability; it is not clear how it may be applied to quenches with nontrivial post-quench GGEs 2- No numerics is presented for the actual function of time $c(t)$, the main object of study

Report

In this manuscript, the author defines a sort of effective central charge which applies to certain quantum quenches in integrable models. The quantity is then studied in various models and quenches, and showed to behave in a physically sensible way. It allows one to make the relation between irreversibility of the renormalization group, and irreversibility of quantum quenches.

I think the concepts introduced are interesting, and the analysis is done quite well. The results are quite convincingly agreeing with the general physics that the author discusses.

The main quantity defined and studied, eq 19, has however many drawbacks: it is based on the theory of Calabrese and Cardy for quenches in CFT, which itself is relatively limited. For instance, as the author mentions, the meaning of the effective temperature is unclear in most situations. Also, all this is based on a thermal understanding of the post-quench state, and it has been understood, as the author explains well, that GGEs are rather the correct ensembles to consider. It is not clear how formula 19 can work in cases where nontrivial GGEs are reached post-quench, as in particular there is no clear evidence of a central-charge interpretation (even as a TBA effective central charge) of any object considered e.g. on page 7. These drawbacks are not those of the author's theory, but rather those of the theory of Ref 11 on which it is based. However, I find it difficult to see how one can go beyond. Thus, for instance it is not clear how GGEs can be studied or characterized within this framework as is proposed in the conclusion (what would be the ``effective central charge" of a GGE?).

The idea that irreversibility of RG is related to quench irreversibility is interesting, although, again, since the quantity proposed has, it seems, limited applicability, then also this connection is limited to certain situations. Note also that there are very general results for irreversibility of quantum quenches (basically having to do with 2nd law of thermodynamics), see for instance the discussion in Takashi Mori, Extensive increase of entropy in quantum quench, J. Phys. A 49, 444003 (2016).

This being said, the author considers only situations where CFT and its thermal interpretation are applicable post-quench (that is, $m_0\gg m$), and thus everything works well, and the RG interpretation makes it interesting - for instance, it makes it clear why it is not possible to do a quench displaying IM $\to$ TIM properties.

The paper can be accepted for publication, but, besides adding the appropriate citations concerning irreversibility of quantum quenches, I'd like to see two things done:

1) There are figures displaying the oscillatory behaviour of integrands, to make the point that under integration these give zero. However, there is no numerical results for the function $c(t)$ itself! It would be very interesting to have graphs of this function, as evaluated using the TBA formalism.

2) It would be very good if the author can give us a more precise idea of how the effective central charge consider is defined when the post-quench is a nontrivial GGE, and why it is a good quantity (e.g. how it relates to a CFT central charge, and thus to RG irreversibility). If there is no clear idea there, then I would suggest making the statements less strong as to the applicability of the present theory.

Requested changes

1) There are figures displaying the oscillatory behaviour of integrands, to make the point that under integration these give zero. However, there is no numerical results for the function $c(t)$ itself! It would be very interesting to have graphs of this function, as evaluated using the TBA formalism.

2) It would be very good if the author can give us a more precise idea of how the effective central charge consider is defined when the post-quench is a nontrivial GGE, and why it is a good quantity (e.g. how it relates to a CFT central charge, and thus to RG irreversibility). If there is no clear idea there, then I would suggest making the statements less strong as to the applicability of the present theory.

---

## Round 1 · Referee Report · Anonymous (Referee 2) · 2017-9-22

Strengths

(1) Clear presentation of the results and workings.
(2) A potentially quite interesting quantity is introduced.

Weaknesses

(1) Limited range of applicability
(2) Work is rather speculative.

Report

The author considers quantum quenches in massive deformations of 1+1 dimensional CFTs. For a very particular (fine tuned) choice
of initial state he defines a quantity c(t), which reduces to the effective central charge at times t=0 and t=\infty. The time dependence of c(t) is studied for several examples.

While I find the basic idea of introducing a quantity c(t) that interpolates between the effective central charges in the initial and steady states quite interesting, there are a number of issues with the proposal put forward. Firstly, in my understanding it is rather restrictive as it basically works only in the particular case where the system thermalizes. For the integrable QFTs of interest here this is a highly fine-tuned and unrepresentative situation as one generically needs to deal with generalized Gibbs ensembles. Moreover, it is not clear to me how to generalize the construction of c(t) to deal with the generic GGE case. However, given that the case considered is the simplest possible one it nevertheless makes sense to consider it first.
I am also unclear about what the interpretation of c(t) is away from its limiting values at t=0 and t=\infty. I suppose the fact that it is generally oscillating in time should make possible interpretations in terms of RG flows difficult if not impossible?
More generally, the connection of the proposed c(t) to RG flows and the utility of knowing the behavior of c(t) are not clear to me.

A couple of minor comments: the reference to the thermodynamic limit above eqn (15) is confusing. Supposedly the authors has in mind that \tau_0,\tau\ll L. I also think that Cardy's work on the return amplitude in the context of quantum revivals should be cited. When discussing the choice of initial state the papers by Cardy and by Mandal et al and possibly others should be cited, as they deal with more generic situations.

Requested changes

(1) I think it would be useful to explain more clearly in precisely what sense the proposed c(t) can be interpreted in terms of RG flows.
To me the connection appears to be somewhat tenuous.
(2) The GE vs GGE issue should be addressed clearly and prominently at an early stage (e.g. when the initial state is introduced), and it should be stated that the current investigation focusses only on the simplest case in which the system thermalizes.

---

## Round 2 · Referee Report · Anonymous (Referee 1) · 2017-12-14

Report

I think the referee has addressed the points correctly. The discussion now puts the results in a better way into the general context. The ideas are interesting. Also I think the author for correcting the mistake found - it is indeed often important to average out oscillations that occur at large times.

---

## Round 2 · Referee Report · Anonymous (Referee 2) · 2018-1-2

Report

Second Report on "Time evolution of effective central charge and RG
irreversibility after a quantum quench"

I think that some of the main points of my first report have not been
addressed in a satisfactory way. In particular it remains unclear what
the connection of the proposed c(t) to RG flows is, and what useful
information can be extracted from the knowledge of c(t).

I therefore do not think that the paper can be published in its
current form. I would suggest the author to rewrite the manuscript
and focus on the calculations he has done, but drop unsubstantiated
speculation (except perhaps in the conclusions).

My more detailed comments and requested changes are listed below.

Requested changes

  1. "RG irreversibility" should be dropped from the title and elsewhere in the manuscript. As far as I can see the paper contains no information on the relation to RG flows and irreversiblity. Sentences like "In particular, we are interested in the consequences of the irreversibility of RG flow in the non-equilibrium evolution" or "The irreversibility of RG flow is reflected in the fact that these quenches can only be performed in one direction..." are as far as I can see inappropriate. If the author wants to speculate on the relation of his results to RG flows and irreversibility he could do so in the conclusions, but clearly identify such comments as speculations.

  2. As I noted in my previous report, the steady state after quenches to an integrable theory is always described by a GGE and only reduces to the Calabrese-Cardy initial state in an (unphysical) limit. For the case of the Ising field theory this was shown explicitly in P. Calabrese et al, J. Stat. Mech. (2012) P07022 section 2.5 and a comprehensive discussion in a CFT setting was given by J. Cardy in Ref. [13]. As I said in my previous report, as a first step it is entirely reasonable to work with the CC initial state here, but the author should clearly state that this is an approximation: for certain quenches the steady state is approximately thermal in the sense that the GGE is very close to a GE (in the free case the mode-dependent temperatures then become almost equal). I take objection to the author's reply to my first report that

"While other more exotic quench protocols might require a modification to the Calabrese-Cardy initial state redefinition of the effective central charge, we feel our proposal describes a wide enough range of of physically relevant quantum quenches."

The fact is that generic quenches require a modification of the CC initial state. The author's proposal applies to an approximate description of an interesting class of quantum quenches.

When introducing the CC state in (15) a discussion along these lines should be added.

  1. I believe the discussion below (12) to be misleading. The return amplitude corresponds to the partition function of a Stat. Mech. system with boundaries, where an analytic continuation in system size rather than temperature has been carried out (the inverse temperature of the Stat. Mech. problems maps onto the system size of the quench problem). This correspondence follows from Fig. 1a by considering the two different transfer directions.

The strip geometry shown in Fig. 1b corresponds to the situation found for CFTs where analytic continuation in time has been performed and an appropriate regularization procedure (that leads to a finite width of the slab) has been employed. In my understanding these are two entirely separate issues, and the first correspondence holds true much more generally than the second.

I think the discussion around eqns (13) and (14) follows from the first correspondence (the system size in the Stat Mech problem maps onto imaginary time in the quench setting).

  1. After (21) there is a discussion that suggests that thermalization can be expected after e.g. mass quenches. This should be changed, cf. my comment (2) above.

  2. Below (27) the author refers to a regime m_0\gg m when he actually means the limit \lim_{(m_0/m)\to\infty}. It is only in this (unphysical) limit where thermalization occurs. For m_0\gg m the system thermalizes approximately. I suggest to refer to approximate thermalization here.

  3. Appropriate axis labels should be introduced in Fig. 5.

  4. The Conclusions section should be rewritten. As the paper does not establish any connection to RG flows, comments speculating on such connections should be phrased much more carefully than is currently done. Fig. 2 shows that the introduced effective central charge is still oscillatory in time, which seems to constitute a problem with regards to notions of irreversibility. I also think that when discussing the notion of irreversibility it would be useful to have a clear and explicit discussion of what precisely the author means by this term in the quench context, as some works in the literature (e.g. arXiv:1711.00015) ascribe a rather different meaning to it.

The "RG" in the sentence stating that an interpretation of the effective central charge at finite times is currently missing should be removed: as far as I can see there presently is no physical or other interpretation of this object.

---

## Round 2 · Author Response

I will first thank the referees for their comments and suggestions which I find quite insightful and reasonable. I believe in this resubmission I addressed all of the issues the referees raised on the previous version, so I hope it is more suitable for publication.

I will point out that the most important modification in this version, which took some time to rewrite, is that I corrected a significant error that I discovered after the first submission. In this version I have a slight modification on the definition of the time-dependent effective central charge, after I realized the previous definition was problematic in some cases. The arising issues were resolved by modifying the definition with an additional time-averaging step which is described in the text (Eq. 21).

I expanded the discussion about the initial states of CFT quenches, and the range of validity of our proposal, addressing an issue that was raised by both referees. The starting point for the proposal are the Calabrese-Cardy initial states, which are known to lead to effective thermalization at late times, and not to a nontrivial GGE. Despite the simplicity of these states, these are known to describe accurately massive-theory-to-CFT quenches, which is the motivation to consider them. We later consider quantum quenches into non-conformal theories and propose a definition for effective central charge based on the general form of the return amplitude, by comparing it to a CFT quench (with Calabrese-Cardy initial states). Our proposal is then expected to be valid and reasonable for quantum quenches which reduce to the Calabrese-Cardy quench in an appropriate limit (taking the post-quench massive parameter to zero). If when we reduce the ratio $m/m_0$, we see that the late-time dynamics become thermal, as is expected in the CFT-limit, then we expect our proposal to be valid. As we discuss in the text, this is enough to describe a reasonably wide range of quantum quenches of massive theories.

While other more exotic quench protocols might require a modification to the Calabrese-Cardy initial state, which would result on a redefinition of the effective central charge, we feel our proposal describes a wide enough range of of physically relevant quantum quenches. It would be interesting to study extensions to other more possible quenches in the future, but this is beyond the scope of this first article on the subject.

As suggested by Referee 1, I have added a plot for the time evolution of the effective central charge, where one can observe the behavior we propose. Namely, the central charge seems to interpolate between the expected values at $t=0$ and $t\to\infty$. At finite times the charge can oscillate as it reaches its asymptotic value. Any such numerical evaluation is, however, limited to short times, as the numerical precision is affected at late times.

Both referees raised the issue of the definition and interpretation of the central charge at late times, when the state is described by a GGE. I have added some discussion on this subject in the text. This interpretation is perhaps clarified by understanding the GGE as a thermal-like state, where each momentum mode can have a different temperature (with the typical constraint that the zero-momentum mode has an equal or higher temperature than the other modes). Given some momentum-dependent temperature, one can then easily define and understand an effective central charge in direct analogy to the thermal one. This GGE central charge has to be smaller or equal to the thermal charge at the same effective temperature (since all momentum-modes then have the same constant temperature, which is the highest value corresponding to the zero-mode temperature). It can be clearly seen that the GGE central charge reduces to the thermal charge in the large-quench limit, where the effective temperature acquires a constant value.

Responding to issues raised by Referee 2, I did not include a deep discussion on revivals, as indeed I am considering the limit where system size, L, is much larger than the length scales given by the effective temperature. The CFT partition function we present is computed on an infinite strip geometry, while revival effects are seen by considering a finite ring geometry. I have not added a discussion of the effects of revivals, since this is presently beyond the scope of my proposal.

Finally, responding to Referee 2, to avoid the risk of too much speculation, I currently do not offer a full interpretation of the oscillating finite-times dynamics of the effective central charge, and its RG interpretation. Studying the time evolution, as in Figure 2b, it is very tempting to think this quantity somewhat describes how the system evolves from one value of central charge to the other, and provides a description of more exotic configurations that are visited in the finite time non-equilibrium evolution. Aside from a few comments in the Conclusions, I refrain from making any strong statements about the finite time evolution, as this would require perhaps a deeper numerical study in a wider variety of scenarios. The only interpretation I can offer in all honesty is that between the relation of the $t=0$ and $t\to\infty$ values of the charge, which is discussed at length in the paper.

---

## Round 2 · List of Changes

-The most significant modification is a change in the definition of the effective central charge (introducing an additional time-averaging step), which fixes some issues with the previous definition.

-Discussion about the Calabrese-Cardy initial states and the range of validity of our proposal is largely enhanced. The issue is now addressed at length in the introduction, an in the section about CFT quenches.

-Discussion was added on the interpretation of an effective central charge corresponding to a GGE state, particularly in Section 6, concerning the free massive boson. The discussion of the GGE vs GE was also enhanced in the introduction.

-A numerical plot was added showing explicitly the time evolution of the effective central charge for the free massive boson quench.

-References suggested by the referees were added, as well as brief corresponding discussions. Other smaller typos were corrected.

---

## Round 3 · Author Response

I thank the referees for their time and work reviewing this manuscript. Attached is the newest version of this manuscript, which I hope addresses the issues that have been raised.

First, it seems Referee 1 was satisfied with the previous version and no changes were suggested. I am happy to see that in his/her opinion the quality of the manuscript has already improved, and I thank again the referee for their work.

In this version I have incorporated several changes, in attempt to answer for some of the issues raised by Referee 2. Here I will review the modifications step by step, corresponding to the items enumerated in the referee's report (see the list of changes below).

---

## Round 3 · List of Changes

1) I have made many changes throughout the paper regarding the discussion of "RG irreversibility", including several mentions in the introduction and the conclusion sections. The main theme has been to soften the tone of some claims. When the value of the effective central charge exhibits some apparent irreversibility properties, c_{\rm eff}(\infty)\geq c_{\rm eff}(0), it is commented that this is suggestive and may point to a connection with the concept of RG flow and irreversibility, by comparing to the known properties of the equilibrium c-function. This is in contrast with previous versions where it may seem like I claim "this IS RG irreversibility", and made it sound like a claim of a rigorous proof.

Accordingly, I have modified the title with the phrase "signatures of RG irreversibility", suggesting that this is not a direct proof of RG irreversibility, but that arguments are presented to suggest what we see may be connected with it. I have opted not to entirely remove "RG irreversibility" from the title, since I feel that would be slightly dishonest to the paper, since this is a subject that is widely discussed, and a central theme in the paper. I hope that this softening the tone of the claims, and the title, will be enough to reflect the nature of what is actually discussed in the paper.

2)The discussion regarding the Calabrese-Cardy initial states has been enhanced. I attempt to make it more clear that this is indeed just a simplified model that is only useful as a first step. A new paragraph augmenting this discussion was added after Eq. 15, just after introducing these states. There is also more discussion added on this at the end of the same section. I have also added in the conclusions section a brief discussion of how one could a attempt a generalization to other kinds of initial states, which do not lead to effective thermalization. This should make it clear that it should be conceptually possible to define similarly an effective central charge for wider classes of quantum quenches, the only limitation is a practical computational one.

3)The discussion of the relation between the return amplitude and the partition function on the strip is clarified. Indeed I agree, that the roles of the space and time coordinates are exchanged in this analogy, and this was not very clear in my previous discussion. This discussion has been modified, and hopefully the relation is now clear.

4) Together with the modifications in the discussion of CC initial states, the perhaps misleading claim has been corrected, along with a more careful discussion of when thermalization is expected to occur.

5) The issue of the limits and conditions which lead effective thermalization has been more carefully addressed, including the limit \lim_{(m_0/m)\to\infty}, throughout the paper.

6)The labels in the plots have been modified.

7) There have been several modifications in the conclusions. Any claims regarding RG irreversibility have been softened. It is explained that the results are suggestive at some connection with this concept, given how the effective central charge is related to the known equilibrium c-function, but any hard claims that make it seem like we have found some rigorous proof are dropped. The discussion regarding the finite-time values of the central charge is also enhanced, making it clear that it is difficult to see if and how it may be connected with RG flow, and making it clear that for now one can only speculate about this. As mentioned before, the discussion of initial states was also enhanced in the conclusions.

---

## Editorial Decision

published